# Starch Nanoparticles: Preparation, Properties and Applications

**DOI:** 10.3390/polym15051167

**Published:** 2023-02-25

**Authors:** Herlina Marta, Dina Intan Rizki, Efri Mardawati, Mohamad Djali, Masita Mohammad, Yana Cahyana

**Affiliations:** 1Department of Food Technology, Universitas Padjadjaran, Bandung 45363, Indonesia; 2Research Collaboration Center for Biomass and Biorefinery between BRIN and Universitas Padjadjaran, Bandung 45363, Indonesia; 3Department of Agroindustrial Technology, Universitas Padjadjaran, Bandung 45363, Indonesia; 4Solar Energy Research Institute (SERI), Universitas Kebangsaan Malaysia, Bangi 43600, Selangor, Malaysia

**Keywords:** starch nanoparticle, preparation, properties, applications

## Abstract

Starch as a natural polymer is abundant and widely used in various industries around the world. In general, the preparation methods for starch nanoparticles (SNPs) can be classified into ‘top-down’ and ‘bottom-up’ methods. SNPs can be produced in smaller sizes and used to improve the functional properties of starch. Thus, they are considered for the various opportunities to improve the quality of product development with starch. This literature study presents information and reviews regarding SNPs, their general preparation methods, characteristics of the resulting SNPs and their applications, especially in food systems, such as Pickering emulsion, bioplastic filler, antimicrobial agent, fat replacer and encapsulating agent. The aspects related to the properties of SNPs and information on the extent of their utilisation are reviewed in this study. The findings can be utilised and encouraged by other researchers to develop and expand the applications of SNPs.

## 1. Introduction

Starch is a native polymer that is abundantly available and widely used worldwide [1,2]. Starch is a carbohydrate. Starch granules are a source of carbohydrates and renewable substances produced by plants in the form of granular constituents of plant parts, such as seeds, tubers and fruits, as sources of stored energy; the shape and nature of starch depend on the botanical source, climate and location from where it is isolated [3,4,5,6,7]. Globally, starch production is mainly based on four raw materials, namely, corn, cassava, wheat and potatoes, with >75% of starch produced from corn [8,9]. 

Starch is a mixture of two macromolecules, namely amylose with linear chains of glucose molecules connected by α-1,4 glucosidic bonds, and amylopectin with branched chains consisting of short amylose groups connected by α-1,6 glucosidic bonds [10,11]. In general, starch contains about 20–30% amylose and 70–80% amylopectin, depending on the source [12]. The ratio of amylose/amylopectin content will affect the functional properties of starch, such as gelatinisation, viscosity and gel stability [13]. Starch has been widely used as an additive in the food industry and other industrial applications, such as pharmaceuticals, drug delivery and composites, because of its low cost and easy availability [14,15]. In the food industry, it is generally used as a thickener and an auxiliary to improve food texture and can be utilized to manufacture sauces, soups and puddings [16,17]. Native starch directly extracted from plants generally has limited industrial-scale applications. It has low thermal stability and resistance to external factors during storage, high brittleness and a hydrophilic nature [18]. Therefore, additional treatment, either physically, chemically or enzymatically, is needed to change the properties of native starch to overcome its limitations [16,19,20,21,22,23,24,25].

For decades, the food industry has attempted to improve the physicochemical properties of starch to ensure the final quality of food products [26]. Several starch studies have focused on particle size and understanding the relationship between nano- and microscopic properties of different materials, where the resulting particles are small and have a large surface area [16,27]. This new alternative, the modification of starch macromolecules from micro- to nanoscale, produces nanostarches, which have attracted considerable attention due to their unique properties, and not only change the particle size, but also enhances their functional properties [6,26]. Nanostarches are of two types: starch nanocrystals (SNCs) are crystalline portions resulting from the disruption of amorphous domains in starch granules, and starch nanoparticles (SNPs) generated from gelatinised starch, which may include amorphous regions [27,28]. SNPs are considered promising biomaterials and offer various opportunities for quality improvement for the innovative development of competitive products [29,30]. SNPs have been widely used in active food packaging [31], bioactive compound encapsulation [32], nanocomposite film [33] and Pickering emulsion stabilisers [11,34].

This review focuses on SNPs, their various preparation methods, including physical, chemical and enzymatic treatments, and the resulting characteristic changes. Characterisation of SNPs in terms of size distribution, crystal structure and various properties compared with native starch and the extent to which SNPs have been utilised were also reviewed. The scheme in Figure 1 summarizes the preparation, properties and application of SNPs.

## 2. Starch Nanoparticles

SNPs are produced through a nanotechnology process that produces nanoparticles with a size smaller than 1000 nm, but are larger than a single molecule [6,35]. Furthermore, Le Corre et al. [36] reported that the morphology and properties of the resulting nanoparticles depend on the botanical source. Some previous studies on SNPs, such as those from potatoes [6], bananas [37], water chestnuts [38], sago [39] and maize starches [40], have been widely carried out.

In general, the preparation of SNPs can be classified into ‘top-down’ and ‘bottom-up’ methods [41]. On the other hand, top-down methods, such as ultrasonication, homogenisation, gamma radiation, acid hydrolysis and others, involve the production of nanoparticles from the breakdown of large particles into small ones based on structural fragmentation using mechanical and chemical forces [30,42]. On the other hand, bottom-up methods produce nanoparticles using a thermodynamic process of controlled molecular assembly, such as nanoprecipitation or self-assembly [42]. Top-down approach technology is the most commonly used method to produce nanoparticles [10].

Top-down methods have the advantage of easy usage, but they are ineffective for the production of particles with the right size and shape [43]. Meanwhile, bottom-up methods produce nanoparticles with controllable shape and size, high yields and short duration times; they require specific chemical reagents and advanced equipment [26,44]. The main considerations in selecting the preparation method of nanoparticles may be related to the size of the resulting particles and scale of production. According to Pattekari et al. [45], top-down methods are more efficient on a larger scale and produce SNPs with a large particle size, whereas bottom-up methods can produce smaller particles, but are more suitable for use on a laboratory scale given the lower resulting yield. 

SNPs have various distinctive properties. They are widely used compared to native starch because of their nanometric size; thus, starch has received considerable attention due to its large surface area per mass ratio and effectively increased interactions [38]. Based on several studies, SNPs are a promising alternative for the formation of stable emulsions, carriers of bioactive compounds and effective packaging developments for the production of food-grade films with great resistance to mechanical effects [35,38].

## 3. Preparation Method of Starch Nanoparticles

SNPs are obtained from the breakdown of granules produced by various methods. The preparation of nanoparticles can be classified based on the preparation method: physical, chemical, enzymatic methods or their combination. Table 1 presents the various methods for the preparation of SNPs from various starch sources.

### 3.1. Physical Methods

Various SNP preparations, such as gamma irradiation, high-pressure homogenisation (HPH) and ultrasonication, have been carried out. These methods are less complicated and less expensive, and can be used to reduce the use of chemicals to prevent leaving residues in the final product [48,67]. Furthermore, physical methods are less-time consuming than chemical methods. However, some physical methods have the disadvantage of being more energy consuming.

Physical preparation using gamma irradiation as an immediate modification technique that causes depolymerization by breaking glycosidic bonds and hydrolysing chemical bonds, and it results in the production of small starch fragments [41]. The preparation of SNPs with this method has been tested on various starch sources, such as cassava and waxy corn starches, with a general application of a 20 kGy dose and a resulting particle size of 20–50 nm [46,47]. 

Furthermore, HPH operates at high speed and shear rate of product flow [68]. Ahmad et al. [48] conducted HPH at a pressure of 250 MPa and reported that the repeated homogenisation process can result in significant size reduction. The preparation of nanoparticles using HPH has been tested on sago and high-amylose corn starches [48,49].

Ultrasonication has been widely used to prepare SNPs, such as those from corn, cassava, quinoa and corn starches [50,51,52,54]. Ultrasonic treatment is promising due to its high yield, being rapid and relatively simple without any purification steps. In the ultrasound modification, sound waves with frequencies higher than the threshold of the human hearing range (>16 kHz) are used. The sound waves generated by ultrasound generate mechanical energy disrupt the starch molecules, causing them to break apart into smaller particles. The mechanical energy also creates microscopic bubbles in the solution. These bubbles rapidly collapse, producing high pressure and temperature, which can also contribute to the breaking of starch molecules into smaller particles. In addition, the mechanical energy of the ultrasound also causes hydrodynamic stress in the solution, which can contribute to the formation of small particles [54,68,69,70]. SNPs preparation by ultrasonication uses a variety of powers, temperatures and times; in general, numerous studies have used 24 kHz power for 75 min, with the increased ultrasonication treatment time resulting in a decreased particle size [50,51]. Among various physical methods for obtaining SNPs, ultrasonication is more advantageous because of the more optimal yield produced than that obtained when using other physical methods. The yield produced using the ultrasonication method is close to 100%, higher than the acid hydrolysis process because the acid hydrolysis partially dissolves the material; of course, this method is very promising because of the high yields, it does not use chemical reagents, and it is faster and relatively simple, without purification steps [51,54]. 

### 3.2. Chemical Methods

The preparation of SNPs using chemical hydrolysis has been extensively studied. According to Wang and Copeland [71], the first acid hydrolysis, which was applied by Nageli, used sulfuric acid, whereas Lintner used hydrochloric acid as a solvent; at the end of the 19th Century, both methods were commercialised to treat starch granules and were carried out under gelatinisation temperature for a certain period to produce acid-hydrolysed starch. The preparation of SNPs by acid hydrolysis considers several factors, such as starch concentration, acid concentration, temperature, agitation and hydrolysis duration [72]. 

Table 1 shows the general chemical preparation of SNPs. Hydrolysis of SNPs generally uses optimal conditions, including 3.16 M H_2_SO_4_ or 2.2 M HCl and a 35–40 °C temperature for 12 h to several days. Dularia et al. [38] reported that the preparation of water chestnut SNPs by acid hydrolysis using 3.16 M H_2_SO_4_ for 7 days at 40 °C produced 27.5% SNPs. Another study reported that the use of sulfuric acid in mungbean starch resulted in a 32.2% yield of SNPs [55]. Maryam et al. [39] used 2.2 M HCl and produced a high yield of 80% in sago starch. Angellier et al. [73] reported that the hydrolysis yield was lower when using H_2_SO_4_ compared with HCl, with a lower time for SNP production; however, their method showed a more stable final suspension with H_2_SO_4_ due to the presence of sulphate groups on the surface of SNPs.

### 3.3. Enzymatic Methods

The enzymatic preparation of SNPs involves hydrolysis using enzymes. The commonly used enzymes include α-amylase, glucoamylase and pullulanase [72]. According to Qiu et al. [27], the enzymatic method is the most efficient for starch degradation. The most important thing in the conversion of starch by enzymatic hydrolysis is that the pullunase enzyme can rapidly hydrolyze α-1,6-glycosidic bonds, releasing a mixture of linear short-chain glucose units from the parent molecule amylopectin, and for hydrolysis treatment by α -amylase, which causes random cleavage of α-1,4 -glycosidic bonds in amylose and amylopectin chains. This enzymatic hydrolysis results in cracks and erosion of the starch granules, resulting in a reduction in the size of the starch particles with the right degree of enzymatic hydrolysis. [63,74,75]. 

Kim et al. [74] reported that waxy rice starch hydrolysed using amylase had a large size of 500 nm and an irregular shape. Irregularly shaped SNPs produced by enzymatic hydrolysis are also found in waxy maize and lotus seed [62,75]. No study has reported the yield of SNPs produced by the enzymatic hydrolysis method, probably due to the limitations of research using this single procedure. However, SNPs preparation has been carried out using combined methods with enzymes, such as enzyme hydrolysis–ultrasonication [62] and enzyme hydrolysis–recrystallisation [40,64,76].

### 3.4. Nanoprecipitation

Nanoprecipitation, as a simple and the most commonly used technique to produce SNPs, is carried out by the gradual addition of aqueous polymer solutions or successive additions of nonsolvents to the polymer solution, which leads to the formation of nanoscale particles [41,61]. For SNP preparation by nanoprecipitation, the starch molecular chain must be completely dispersed in the solvent beforehand, and the process is mainly based on the deposition of the biopolymer interface and displacement of water-miscible semi-polar solvents from lipophilic solutions [61,77]. SNPs are usually prepared by the precipitation of a starch paste solution using ethanol, propanol, isopropanol or butanol [27]. According to Tan et al. [78], nanoprecipitation requires high levels of non-solvents, such as acetone, ethanol or isopropanol, which will inhibit the production and application of SNPs. Some previous studies have reported the nanoprecipitation preparation of SNPs from various starch sources (Table 1).

Wu et al. [77] reported that the particle size of SNPs produced is influenced by the proportion of non-solvent used in the nanoprecipitation method; that is, the particle size decreases when the proportion of non-solvent increases. Qin et al. [6] observed that the amylose–amylopectin ratio affects the characteristics of the resulting SNPs. Butanol can only form a complex and precipitate with amylose, but not with amylopectin. The processing of starch into shorter and more crystalline amylose via acid hydrolysis (lintnerisation) or acid-alcohol hydrolysis is required for the production of nano-sized particles by butanol complex [79,80]. The authors reported that the higher the amylose content of native starch, the higher the relative crystallinity of SNPs, and the resulting V-type diffraction pattern is derived from the single helical structure of the inclusion complex consisting of amylose and ethanol [6]. Winarti et al. [65] reported that SNPs from arrowroot starch produced by nanoprecipitation using butanol reached a yield of 20.65–23.8%.

### 3.5. Combined Methods

SNPs can also be prepared by combining various preparation methods. The combination of methods is intended to produce better SNP properties compared with those obtained using a single method. As shown in Table 1, the combined method of acid hydrolysis–nanoprecipitation using sago starch was reported by Maryam et al. [39]. This combined method can produce SNPs with smaller sizes compared with those obtained using acid hydrolysis alone. The authors reported that the addition of precipitation treatment for 12 h can provide a minimum particle size compared with that 24 h. This production was carried out based on the properties of amylose, which can form inclusion complexes with ethanol and n-butanol; the process triggered the formation of a single left helical structure as a result of the rearrangement of the starch structure, which was gelatinised. Acid hydrolysis–nanoprecipitation with ethanol produces a higher yield than precipitation with butanol [39].

Kim et al. [60] reported that the combined process of acid hydrolysis–ultrasonication using waxy corn starch and acid hydrolysis using H_2_SO_4_ for 6 days at low temperature (4 °C) can produce a hydrolysate that is resistant to ultrasonication treatment. Ultrasonication effectively broke down the starch hydrolysate produced. Previously turned into nanoparticles, the yield reached 78% with a particle size of 50–90 nm. The combined treatment of acid hydrolysis–ultrasonication can degrade lotus seed nanoparticles, which showed a significant effect on the resulting size; the increase in ultrasonic power produced high crystallinity with small particle sizes by weakening the interaction of starch molecules and destroying the amorphous regions [62]. Several studies have mentioned enzymatic hydrolysis–recrystallisation; as reported by Qin et al. [6], hydrolysis with pullulanase followed by recrystallisation at 4 °C on waxy corn starch increased crystallinity, produced SNPs with particle sizes of 60–120 nm and attained high yields above 85% compared with conventional hydrolysis. Enzymatic hydrolysis–recrystallisation of elephant foot yam starch yields 56.66–61.33% [64].

## 4. Properties of Starch Nanoparticles

### 4.1. Amylose Content

Amylose content can affect the physicochemical properties of starch, such as gel formation and adhesion. The amylose content may vary depending on the botanical source of starch granules [16,81]. The development of SNPs can result in differences in amylose content. Torres et al. [58] reported a drastic decrease in the amylose content of acid-hydrolysed nanoparticles in Andean potato starch by up to 80% compared with native starch; in addition, the combination treatment of hydrolysed SNPs with ultrasonication showed a decrease in amylose content. The decrease in amylose content can be attributed to hydrolysis, which eroded the amorphous regions consisting of amylose molecules, compared with crystalline regions of starch granules which are generally more resistant to hydrolysis [30]. The amylose and amylopectin ratio, the inter-chain organization and the type of crystallinity pattern play a significant role in NSPs properties. Furthermore, Bajer [82] reported that the influence of amylose content on nano-starch was crucial. 

The preparation of arrowroot SNPs by nanoprecipitation with 5% butanol can increase the amylose content; the amylose content will increase by precipitation with alcohol because only amylose forms complexes with these alcohols. However, the results showed that the amylose content was not significantly different from that obtained with treatment using butanol at a concentration of 10% [65]. In addition, nanoparticles can be produced by physical preparation, such as HPH. This method triggers the destruction of the starch structure by reorganising the crystal region outside the granule, which can free the amylose, thereby increasing the amylose ratio [49]. Based on the work of Apostolidis and Mandala [49], the amylose content of corn starch obtained with this treatment can be influenced by two factors (pressure/cycle). The higher the pressure and the more cycles applied to this method, the greater the effect on the treated sample, where repeated treatments produce new structural domains from amylose leaching.

### 4.2. Particle Morphology and Size Distribution

The morphology, structure and size distribution of SNPs can be characterised through several testing techniques, such as atomic force microscopy, scanning electron microscopy and transmission electron microscopy by controlling the preparation conditions [68]. The morphology of SNPs may differ depending upon the botanical sources and preparation methods [81]. Table 2 presents a variety of starch sources, preparation methods and observations on the morphology and size of the tested SNPs.

Based on various studies, SNPs can be round, flat, platelet, ellipsoidal or irregular with cracked and porous surfaces. The differences in morphology depend on the botanical source and preparation technique or modification of SNPs. The size distribution of different particles also varies depending on the starch source, where the smaller the starch granules, the smaller the nanoparticle scale produced [6,58,68]. A smaller nanoparticle size produces different functional characteristics from standard particle size, which has led to their use in various industrial developments [30]. Furthermore, amylose content can affect SNPs, with high amylose content producing large nanoparticles [30].

Physical preparation methods for SNPs can be carried out with various treatments. Numerous studies have used ultrasonication on cassava starch [52], quinoa and corn starch [54], corn starch [53] and waxy corn starch [50,51], which resulted in the size distribution of molecules and different morphological forms depending on the starch source and the time and frequency of sonication treatment. Remanan and Zhu [54] reported that ultrasonication significantly disrupted the starch granule structure and crystalline properties of starch. The reduction of granules to SNPs is influenced by the solvent composition used in this method, such as water content, which is a prerequisite for destroying starch size [53]. Other studies have used gamma irradiation at a dose of 20 kGy on cassava SNPs, and caused the formation of laminar aggregates with a large specific surface area; numerous OH groups on the surface were connected by hydrogen bonds, with a resulting particle size of 30–50 nm [46,47]. Physical treatment can be carried out using HPH. Apostolidis and Mandala [49] reported an increase in pressure, and the homogenisation cycle led to a more considerable size reduction, which was effective for starch breakdown. These results are also supported by research conducted on sago starch, where the treatment of five cycles of HPH resulted in a significant reduction in size [29].

Based on the research by Jeong and Shin [59], granule size decreased along with the increased days of acid hydrolysis treatment. Kim et al. [56] reported that the nanoparticles produced in hydrolysis using acid caused erosion of the amorphous lamellae and the release of nanocrystal components. During this treatment, the molecular distribution expanded as the hydrolysis time progressed due to surface erosion fragmentation during stirring, the hydrolysis process and agglomeration, which resulted in irregular shapes [83,85]. The resulting nano starch is diverse with different morphological characteristics of starch, which can be attributed to the biological origin and physiology of plant biochemistry [55]. 

Based on several studies that carried out the nanoprecipitation method on several starch sources, most of the SNPs are spherical/elliptical/irregular, and others exhibit slight aggression; SNPs occasionally show an uneven distribution with an average distribution range of 30 nm to >250 nm [6,37,40,65,66]. The resulting differences occur due to differences in starch sources, time and precipitation reagents used. Maryam et al. [39] combined the acid hydrolysis method with the precipitation of sago starch. They produced a substantially smaller molecular distribution than the acid hydrolysis treatment alone. In the precipitation process with hydrophobic components, such as butanol and ethanol, when amylose accommodates hydrophobic molecules, it will form a textured single helical crystal due to the rearrangement of the gelatinised starch structure.

According to Foresti et al. [75], hydrolysis using enzymes in starch occurs through three stages, namely, diffusion to the solid surface and adsorption until catalysis. The increase in hydrolysis time increases the fraction of fragmented particles. Grain fragmentation can be evidenced by a progressive reduction of the average grain diameter. Kim et al. [74] reported that enzymatic hydrolysis by amylase causes the starch surface to crack and become porous, which indicates that the enzyme penetrates the granules. They also concluded that the right degree of enzymatic hydrolysis can reduce starch particle size, but excessive hydrolysis can increase it.

### 4.3. Starch Crystallinity

The crystallinity level is the ratio between the mass of crystal domains and the total mass of whole SNPs, the crystallinity of which is mostly ascribed to amylopectin [86,87]. The crystalline structure of starch can be observed using X-ray diffraction (XRD). Using the X-ray diffraction pattern, starch can be classified into several types: A, B, C and V, whereas for the low quality x-ray diffraction pattern, it has about 70% of the starch polymer in an amorphous state [88,89]. The degree of crystallinity varies depending on the starch source and preparation method used: A-types have double helices tightly packed and are commonly found in cereal starches; B-types have a high amylose structure contained in tubers, stems and fruits, the crystalline part of which is formed of six left-handed parallel-stranded double helix packed in a relatively loosely packed hexagonal unit; C-types are considered a mixture of forms A and B and present in leguminous starches; and V-types can be observed during the formation of complexes between amylose and lipids [52,87]. Table 3 presents the crystallinity of SNPs from various preparation methods.

Changes in the crystal structure of starch during the production of SNPs have been investigated, and studies have confirmed that different modifications applied to various starches will affect the category of starch crystallinity. SNPs made using physical treatments, such as gamma irradiation, ultrasonication and HPH, showed a decrease in crystallinity and an amorphous structure. Based on the research by Lamanna et al. [46], decreased crystallinity and amorphous patterns were observed in cassava and waxy maize starches. In the preparation using HPH, crystallinity also decreased due to the increase in homogenisation pressure applied to starch granules, but a crystalline B-type was maintained [90]. Moreover, da Silva et al. [52] reported that ultrasonication affects the crystal structure, which results in severe disruption of the crystal structure of amylopectin and the amorphous character of the resulting SNPs. This result was also observed in the ultrasonication treatment of waxy corn starch, which caused the loss of diffraction peaks; as a result, crystals were lost during the ultrasonication fragmentation process [50,53].

The XRD patterns of all SNPs generated from nanoprecipitation showed V-type crystallinity, which is unrelated to the native crystal type of starch [66]. The V-type crystallinity is derived from the single helical structure of amylose and ethanol. In addition, gelatinisation in this method destroys the crystallinity of A, B and C-types, with most of the relative crystallinity decreasing due to the weak crystallinity intensity of the single helix of SNPs and low number of single helices during the nanoprecipitation process. The number of single helices is low during the nanoprecipitation process [6,40]. Winarti et al. [65] reported that nanoprecipitation of arrowroot using a butanol complex caused an increase in the degree of crystallinity and a shift in the crystal from A to V-type. This shift in crystal type caused a loss in starch integrity during gelatinisation [65,91]. Qin et al. [6] investigated a positive correlation between the amylose content of starch and the relative crystallinity of starch with high amylose content, which resulted in an increased crystallinity in this nanoprecipitation treatment. The degree of crystallisation is influenced by the percentage of amylopectin, chain length, crystal size, orientation of the double helix in the crystal region and the degree of interaction between the double helix [64].

The SNPs obtained by acid hydrolysis of native starch granules have a higher crystallinity than the parent granules [59]. A-type starch is more resistant to acid hydrolysis than B-type crystals, where the crystal structure of B-type starch is more easily disturbed by acid hydrolysis than B-type crystals. In addition, the degree of hydrolysis is not positively related to the change in crystallinity as measured by X-ray analysis [56]. Furthermore, Maryam et al. [39] explained that the hydrolysis period can affect starch content. Starch crystallinity increases after hydrolysis, in which the hydrolysis process does not change the pattern of starch crystallinity, but only changes its crystallinity index. This condition indicates that hydrolysis only destroys the amorphous region to obtain more starch crystals. SNP crystallinity in enzyme-hydrolysed starch causes a relative increase in crystallinity associated with extensive degradation, especially in the amorphous region of starch granules (Foresti et al., 2014). In SNPs made from a combination of enzymatic hydrolysis followed by recrystallisation, the relative crystallinity increases, and a B + V-type crystalline pattern emerges [40,76].

### 4.4. Thermal Properties

The thermal transition behaviour of SNPs has been characterised using differential scanning calorimetry and thermogravimetric analysis. These thermal analyses are critical because they determine the conditions under which the use of SNPs is applied in industry. The resulting behaviour will determine the appropriate processing conditions to produce a final product that will remain stable [15,92]. Table 4 presents the thermal characteristics of SNPs from various studies.

SNPs prepared using physical methods, such as gamma irradiation, show a low drop in degradation temperature and a sudden weight loss. This finding is due to the initiation of SNP degradation at the surface, which has a high amount of hydroxyl groups [46]. da Silva et al. [52] reported a decrease in the degradation temperature of ultrasonicated cassava SNPs, which led to a lower thermal stability compared with that of native starch, in addition to a decrease in gelatinisation temperature caused by the weakening of hydrogen bonds in the amorphous region. According to Zhu [69], gelatinisation-associated SNPs can be affected by starch-type composition and ultrasonication experimental conditions. A decrease in gelatinisation enthalpy on ultrasonicated SNPs was also detected in quinone and waxy maize starches [50,53,54]. After ultrasonication, reductions in crystallinity and melting temperature were obtained due to disruption with starch particles. A significant increase in hydrogen and Van der Waals bonds occurred [48]. The modification of sago starch using high-pressure homogenisation caused a slight increase in the degradation temperature, which indicates an increase in thermal stability of modified starch after the treatment [48]. The combination method of enzyme hydrolysis followed by recrystallisation decreases thermal stability [63].

The thermal properties of SNPs obtained with acid hydrolysis exhibit increased T_p_ and T_c_ with the length of hydrolysis time, but a decrease in T_o_ after acid hydrolysis; the distribution of long-chain amylopectin in waxy rice starch changes to produce nanoparticles [59]. Increased values of T_p_ and T_c_ and decreased T_o_ were also reported in waxy maize, normal maize and unripe plantain fruit [56,83]. The decrease in T_o_ occurs due to the separation of the crystalline region from the unstable amorphous region. An increase in enthalpy was found in this treatment due to the rearrangement of the starch chain with the increase in crystallinity. However, the enthalpy can decrease at prolonged hydrolysis times, where differences in the susceptibility of the crystal region to hydrolysis may depend on the origin and crystal structure of starch [56]. According to Ding et al. [93], the enthalpy of an endothermic reaction reflects the number of crystals and double-helix chains that affect the amylose–amylopectin content and length distribution of amylopectin molecule. The thermal characteristics of various SNPs produced using the nanoprecipitation method, which were studied by Qin et al. [6], showed a decrease in the enthalpy of gelatinisation due to the single helical structure of nano starch being more susceptible to disintegration than native starch. The highest gelatinisation enthalpy and stability were found in high-amylose corn starch because it has a high amylose content and a high density of crystal structure [6].

### 4.5. Functional Properties

Several parameters have been used to examine the functional properties of SNPs. Winarti et al. [65] observed the functional properties of the swelling volume and solubility of arrowroot SNPs produced using the nanoprecipitation method with butanol; they showed increases in swelling volume (5.28–7.92 g/g) and solubility (9.43–16.89%) compared with the native ones. Other studies, such as those on potato and cassava SNPs obtained with mechanical treatment, showed a significant increase in swelling volume [94]. Jeong and Shin [59] reported that waxy rice SNPs prepared by acid hydrolysis method resulted in a decreased water binding capacity. However, no significant change was observed as a function of hydrolysis time.

### 4.6. Digestibility Properties

The digestion of starch granules is a complex process that includes the diffusion of enzymes to the substrate, which affects substrate porosity, the absorption of enzymes in starch-based materials and hydrolytic events. The in vitro digestibility of SNPs increases compared with that of native starch, generally due to the increased surface area of nano-sized starch [95]. Based on the research of Suriya et al. [64], the percentage of SNP digestibility was increased to 41.29–43.24% by debranching with pullulanase followed by recrystallisation for 12–24 h, with high starch digestibility resulting in shorter retrogradation time. According to Ding et al. [93], the digestibility of starch can be affected by the type of starch, particle size, crystallinity, amylose–amylopectin ratio and retrogradation conditions.

Meanwhile, studies on the enzymatic digestibility of arrowroot SNPs obtained by linearised method and butanol precipitation for 24 h caused a reduction in starch digestibility [65]. This finding was also observed in maize SNPs, which showed the lowest level of hydrolysis that can be attributed to the compact structure of SNPs formed during the recrystallisation of short-chain glucans; this condition resulted in an increased number of short chains, which made enzyme digestion more difficult [96]. Studies on the percentage of resistant starch resulting from hydrolysis showed maximal values until day 8, but the numbers drastically decreased on the next day in waxy rice starch [59]. Several studies have shown that a low hydrolysis rate of SNPs results in numerous non-hydrolysed SNPs. By contrast, acid hydrolysis digests recrystallise amorphous amylose to form a new double-helical structure, which is highly crystallised against enzymatic degradation; the formation of these SNPs is accompanied by crystallinity, which increases enzymatic resistance [65,96,97]. According to Oh et al. [98], when the concentration of SNP increases, digestion inhibition increases, in which SNPs change the secondary structure of and potentially inhibit α-amylase; thus, SNPs have the potential to reduce glucose absorption for diabetics.

## 5. Applications 

### 5.1. Pickering Emulsion

Emulsion systems are applied in various fields, such as food, medical, pharmaceutical and cosmetic industries. However, the contact between water and oil is inherently unstable and will break down over time. Starch is considered a promising alternative for emulsion stabilisation because it is an abundant, cheap, non-allergenic material, and its biodegradability meets the increasing consumer demand for plant materials; emulsion stabilisers using starch have been applied in solid stable emulsions known as Pickering emulsion [99]. Pickering emulsion consists of solid particles which have been used to stabilize oil–water systems and monomer/polymers emulsions [100,101]. Pickering emulsions are often considered very stable due to the almost irreversible adsorption of the particulate stabiliser interface [100]. The use of particles can be a new strategy for emulsion stabilisers in food, such as bio-based particles using starch; numerous studies have shown that physical stability can be attributed to different stability mechanisms compared with conventional emulsifiers [102]. Solid particles can generally stabilise oil droplets by adsorption at the interface by accumulation to form a layer with high mechanical strength and overall functional properties, as well as stable Pickering emulsion depending on the oil–water interface [103]. The stabilisation of particles at the interface can affect barrier properties, thickness, charge and interfacial tension, which affect the behaviour of the resulting emulsion [104,105].

Nanostarch as a stabiliser of Pickering emulsion is promising because the emulsifying capability of nano starch can be significantly improved through hydrophobic modification [11]. In addition, the use of nano starch as a Pickering emulsion stabiliser has attracted widespread attention because of its small size, wide surface area, non-toxicity, biocompatibility, biodegradability, low cost and food-grade nature; thus, these starch particles can be used as attractive stabilisers [11,103,106]. Wei et al. [107] reported that the larger the size distribution of nanoparticles, the more stable the Pickering emulsion. Likewise, in a previous study [108], the Pickering emulsion was stable at a high concentration of total nanoparticles with a low oil ratio because a high number of particles were adsorbed to the interface and formed a physical barrier, which resulted in small droplets. Nanoparticles act as emulsion stabilisers by ensuring the homogeneous state of the emulsion. Nanostarch particles used as Pickering emulsion have been extensively studied [11,34,109,110,111,112,113]. Table 5 presents various applications of nano starch to stabilize Pickering emulsions.

Li et al. [114] reported that Pickering emulsions made with the addition of 0.02 wt% nanocrystals can remain stable for more than 2 months without droplet variations and coalescences. Another study showed an increase in emulsion stability up to 28 days with the addition of taro SNPs at 7% concentration. In addition, emulsion stability does not always increase based on nanoparticle concentration because the resulting emulsion droplets can agglomerate and disrupt the structure [115]. Octenyl-succinic anhydride (OSA) SNC presents an increased emulsification capability compared with untreated starch due to the formation of a superficial charge that can increase the repulsion forces between oil droplets in the nanocrystals [116].

SNPs from acid-hydrolysed starch granules using H_2_SO_4_ or HCl can carry a surface charge and can be used as Pickering emulsion stabilisers; the repulsion that occurs in charged nanocrystals can play an important role in influencing the emulsion capability of SNPs in dispersions [117]. H_2_SO_4_-hydrolysed SNCs are negatively charged due to phosphate groups on the surface. When the pH is low, the electrostatic repulsion is reduced, which results in droplet aggregation and an increase in particle size in stabilised emulsion. However, the size of emulsion can be adjusted by changing the pH [117]. Haaj et al. [111] stated that HCl can give a better effect by causing polymer dispersion, which results in a smaller average size.

According to Miao et al. [122], high-branching SNPs play a crucial role in the physical stability of Pickering emulsions. A high-branching structure implies the stiffness of starch, which results in a thick adsorbed layer around the droplets with excellent barrier properties for a long time. Lu et al. [123] also reported that the stability capacity of SNPs depends on the starch source, where normal corn starch showed a better stability compared with high-amylose starch. The concentration of starch used in the emulsion can increase the stability of emulsion associated with particles in the continuous phase, which increases the emulsion viscosity and thereby inhibits the separation of the emulsion phase and aggregation of oil droplets; as a result, the emulsion becomes more stable [118,124,125]. In addition to the concentration of added nanoparticles, emulsion stability can be affected by SNP type, size and hydrophobicity, where a small hydrophobicity contributes to increased emulsion stability [34,99,126]. Figure 2 shows a graphical abstract of SNPs application as Pickering emulsion.

### 5.2. Bioplastic Filler

At present, plastics are increasingly and excessively used, which results in a negative impact on the environment and ecosystems. Therefore, efforts should be exerted to develop more eco-friendly plastics, such as bioplastics composed of biodegradable biopolymers. However, in general, bioplastic products have drawbacks, such as high permeability to water and oxygen, brittleness, low melting points, low mechanical strength, susceptibility to degradation during product storage or use and non-resistance to chemical compounds [127]. 

Bioplastics are produced from the fusion of biopolymers, plasticisers and fillers; additional components in the form of fillers, such as starch, can be used to improve the characteristics of bioplastic packaging [128,129]. The production of biopolymer nanocomposites containing fillers with nano-dimensions is one of the newest methods used to improving the functional properties of biopolymer films; the use of nanofillers in biofilms increases gas and thermal resistance [130]. The incorporation of nano starch from different botanical sources in packaging films as reinforcement components or polymer matrix fillers has been widely studied [96,131,132]. Starch has a good capability to make biodegradable films because of its suitable mechanical properties in bioplastic production [33]. The use of nano starch as a nanofiller can increase the modulus of elasticity and tensile strength and decrease elongation at break and water vapor permeability (WVP) [92]. Table 6 presents several applications of nano starch as fillers in bioplastics.

The addition of SNPs filler can be used as the main feature to obtain effective mechanical and thermal strength in the manufacture of bioplastics, where a structural change occurs in the resulting product. According to Al-Aseebee et al. [140], the mechanical properties of bioplastics are influenced by the interaction between the nanofiller and the matrix because nanoparticles have a large surface ratio. In general, in most cases, the modulus of elasticity and tensile strength increase, which is related to the decreased percentage of elongation in bioplastics produced by the addition of SNPs; similar results can be found in the manufacture of bioplastics using casting methods or other methods, such as extrusion [30,47]. The percentage of elongation related to the difference in stiffness decreases due to the interaction between the matrix and the processing agent [135]. The improvement of mechanical properties of the resulting nanocomposite films can be attributed to the structure and stiffness of nanoparticles, which limit the movement of starch chains [36,141].

In general, the increase in tensile strength with increased concentration of starch nanofillers is considered a result of the strong interactions between the filler and reinforcing matrices. According to Hakke et al. [134], the concentration of nanofillers increases the tensile strength resistance and flexural stress resistance of bioplastics, but the use of fillers decreases by more than 25% due to agglomeration of SNPs, which causes repulsion between the filler and the matrix. The authors also explained that the addition of nanofiller SNPs can induce interactions in the polymer matrix, such as hydrogen and non-covalent bonds; the increase in % elongation with the increase in SNP concentration also results in an increased Young’s modulus of elasticity, which is higher until agglomeration begins, and nanofiller SNPs are distributed. The hydroxyl groups are uniformly connected to the surrounding polymer matrix, which provides additional strength to the applied force [134].

The interaction of filler and matrix in bioplastic packaging forms a strong bond, which causes difficulty for air and water vapor to penetrate through the film [142]. According to Hakke et al. [134], an increase in the concentration of added SNPs will increase film compactness; therefore, the more compact the resulting film, the longer the time required for water vapor and air to penetrate the matrix. The composition of bioplastic constituents used significantly determines the permeability of the resulting packaging; the higher the concentration of filler used, the denser the structure inside the packaging; thus, the packaging pores will be smaller, which causes difficulty for water vapor and air to penetrate the packaging walls [134,143].

Hakke et al. [134] reported a decrease in WVP along with the addition of SNP at a certain concentration; the addition of 20% SNP caused a 60% reduction in WVP; in addition to SNP concentration, temperature affected the WVP produced. According to Mukurumbira et al. [136], the presence of SNCs reduced the water affinity of the film, which formed between the starch matrix and nanocrystals, where the SNCs and starch film exhibited the same polarity; thus, the interaction of SNCs with the water matrix was minimal. In addition, the presence of dispersed SNCs created a winding path for the movement of molecules, and as a result, the longer diffusion path of water molecules affected the reduced permeability [133,135]. However, in the study of Li et al. [133], the addition of SNCs above 5% caused a slight increase in VWP, possibly due to aggregation; thus, SNCs failed to effectively prevent the migration of water molecules. Likewise, Ahmad et al. [48] showed an increase in WVP when the addition of nanoparticles was above 8%; this finding can occur when the use of filler exceeds the maximum concentration. Excess nanoparticles increase the water affinity of starch due to the abundance of hydroxyl groups and the high possibility of agglomeration in the filler [144,145].

According to Mukurumbira et al. [136], the thermal properties of nanocomposites are an important factor in determining suitable processing conditions. The increase in the composite film’s thermal stability indicates the strong interaction between SNCs and the film [136]. An increase in melting temperature was observed with the addition of nanocrystals, and a decrease in film enthalpy was noted with the increased concentration of SNCs to the starch film; this finding may be due to SNCs inhibiting the lateral arrangement of starch chains and the crystallisation of starch films [133,136,143].

Basavegowda and Baek [31] reported that the mechanism of action of SNPs as a filler for increasing stability is the interaction between the filler and the matrix, which forms a barricade barrier that can inhibit the transfer of heat and energy. Hakke et al. [134] reported that the values of T_g_, T_m_ and ∆H increased along with the addition of filler; the increase in ∆H can be associated with the interaction of active nanoparticle starch granules and binding to the packaging matrix; therefore, the increased changes in the polymer results in the increased energy requirement to change the bioplastic polymer; the investigation also showed that increasing the concentration of corn SNPs in the polyurethane solution can increase the cohesiveness of the resulting film. The glass transition temperature and nanocrystals increased, which was associated with the absorption of strong polymer chains on the nanocrystal surface; as a result, the polymer chain matrix bonds were formed [136].

According to Zou et al. [138], T_g_ and T_m_ can be affected in two opposite ways, including movement of the soft segment, which can be suppressed by steric resistance from nanocrystals, and hydrogen bonds on the surface of SNCs, which cause T_g_ and T_m_ to move to a higher temperature. On the other hand, the addition of SNCs may cleave the native interactions of soft and hard segments, which results in changes in the microphase matrix structure, where the soft segments can escape from the binding of hard segments, which causes a decrease in T_g_ and T_m_ [138]. 

### 5.3. Antimicrobial Agent

Raigond et al. [146] reported that incorporated nanoparticles can be used as antimicrobial agents that can improve food safety by minimising the growth of pathogenic microorganisms. Nanoparticles with a large surface area allow more microorganisms to adhere, which increases antimicrobial efficiency. SNPs can be developed for the compartmentalisation of active substances, such as stabilising antimicrobial compounds, which are known for their effectiveness [59,147]. Hakke et al. [134] reported that the addition of 5% SNPs to the manufacture of nanocomposites can attain the maximum reduction in bacterial resistance because the uniform distribution of SNPs in polyurethane solution can reduce the overall pore size of the film. In another study, the addition of SNPs to the stabilisation of potassium sorbate resulted in a retention capacity between 41.5–90 mg/g, which indicates that the added SNPs can be used as antimicrobial agents in food systems [148].

Qin et al. [149] showed that SNPs can significantly increase the antibacterial activity against *S. aureus* and *E. coli* from curcumin. Furthermore, Nieto-Suaza et al. [150] reported the preparation of films with banana starch and aloe vera with the addition of acetate SNPs and curcumin; the authors speculated that the resulting films could control microbial growth by increasing antibacterial activity in food products. Another study on nano starch loaded with carvacrol showed a good antimicrobial activity; that is, a 62% reduction in microbial growth of *E. coli*, 68.0% in *Salmonella typhimurium* and other tested bacteria [151]. Increased antibacterial activity was also found in SNPs loaded with polyherbal drugs [152].

Furthermore, Dai et al. [153] reported that the added nanoparticles can destroy bacterial cell walls and membranes, which results in the antimicrobial effect of bacterial apoptosis. The addition of 0.5 mg/mL starch to antibiotics increased the inhibition zone against *S. pyogenes*. Thus, SNPs can be used to increase the effectiveness of antibiotics [154]. Qin, et al. [155] concluded that branched SNPs obtained by ultrasonication can be used as an antibacterial enhancement factor in encapsulating epigallocatechin gallate (EGCG), especially against *E. coli*.

### 5.4. Fat Replacer

SNPs can be applied as an imitation or substitute for fat in food [92]. Fat substitutes act as imitators of triglycerides but do not replace fat on a gram-for-gram basis [30,41]. The particle size of starch is important in determining organoleptic tastes, such as the taste of fat in the mouth. Small-sized SNPs can be promising fat substitutes, and the mixing of SNPs with other components, such as smooth cream, produced properties similar to those of fats [41]. In addition, fat substitutes will decrease calorie levels [30].

Kaur et al. [156] reported that corn SNPs can replace fat in salad dressing products by up to 60% without reducing their quality characteristics. Another study revealed that using sweet potato SNPs as a substitute for fat in ice cream products allowed fat reduction, which is beneficial for low-fat ice cream production. The authors concluded that the application of SNPs can produce superior-quality products; that is, it significantly improved the texture of ice cream, which gained the approval of panellists [157]. Characteristics of fat substitutes can improve emulsion stabilisation, as shown in the study of Javidi et al. [120]. The authors revealed that fat replacement using corn SNCs resulted in a decreased droplet size and an increased zeta potential, which could be used to produce more hydrogen bonds; thus, the network between droplets formed was substantial. Nano starch as a fat substitute is useful as a stabiliser for oil–water emulsions. Considering its biodegradability, nano starch is promising in the food industry concerning public health [120].

### 5.5. Encapsulating Agent

SNPs can be applied in encapsulation systems, which are an attractive alternative for bioactive compounds [32,35]. The use of nano starch as a superior encapsulation material is due to its biocompatibility, low viscosity at high concentrations, large surface area, non-toxicity, low cost and ideal trapping of bioactive materials [26,86]. Several food ingredients and pharmaceutical application materials have been encapsulated using SNPs [158,159,160,161,162]. Table 7 presents several applications of SNPs as encapsulation agents. 

According to Ahmad and Gani [169], encapsulation efficiency (EE) determines the amount of core material trapped in the carrier material, and the percentage depends on the number of compounds initially loaded during the encapsulation process. The highest percentage of EE in starch without modification treatment was shown in vitamin E (VE), with soluble SNPs reaching 91.63%; this finding also indicated that most of VE can be trapped in SNPs [166]. Based on Table 7, the highest proportion of EE up to above 97% was produced in conjugated linoleic acid (CLA) encapsulation using encapsulating agents from waxy corn OSA nanoparticles; CLA was effectively trapped in nanostructured particles and can be absorbed with the initial modification treatment, which will effectively increase the EE [171]. Likewise, the acetylation of banana SNPs shows a better capacity for curcumin encapsulation than nanoparticles without acetylation [37].

The EEs of various SNPs are different [167]. The highest EE was found in horse chestnuts, and it was caused by the smallest diameter size; small particles have better EE and can form a better film around the core and retain encapsulated molecules [167,173]. Remanan and Zhu [54] reported that a difference in the efficiency of routine encapsulation of quinoa SNPs with corn starch. The smaller particle size of starch quinone nanoparticles with larger specific surface area and stronger adhesion may have contributed to the larger EE, which resulted in better retention of rutin in the encapsulated system [174]. According to Zhu [175], the routine encapsulation of nanoparticles can be caused by the formation of non-inclusion complexes by hydrogen bonds, hydrophobic interactions and electrostatic and ionic interactions. Molecular interactions occur between dissolved rutin and starch chains mostly due to the presence of hydrogen bonds (between the hydroxyl group of rutin with oxygen atoms from starch glycosidic bonds) [176].

Numerous researchers have argued that the difference in EE can be caused by differences in the type of starch and the degree of interaction between starch molecules and bioactive compounds, which can facilitate the incorporation of these compounds in starch networks [54,177]. EE also depends on several other factors, such as the concentration of core material, encapsulation reaction and synthesis process [178]. In general, the application of SNPs as an encapsulating agent can increase the percentage of EE. However, Ahmad and Gani [169] reported the encapsulation of resveratrol and SNPs at a ratio of 1:40, which resulted in a decrease in EE; this result is related to the high ratio of active ingredients, which may not be completely trapped; thus, a decrease in the EE of resveratrol in SNPs was observed. Qin et al. [155] reported that the length of ultrasonic irradiation of unbranched SNPs to be encapsulated caused a gradual decrease in the EE of EGCG, but the combination of irradiation and re-crystallisation techniques improved the EE. EGCG encapsulation using debranched waxy corn with different treatments also increased the EE [172].

## 6. Conclusions and Future Research 

SNPs have been widely studied. They are found in various shapes and sizes based on the starch source and size reduction method used. In general, the preparation of SNPs can be classified into ‘top-down’ and ‘bottom-up’ methods. Starch-produced nanoparticles are used as Pickering emulsions, bioplastic fillers, antimicrobials, fat replacers and encapsulating agents. Thus, these nanoparticles have the potential to be produced on a large scale and further developed into food products.

The development of starch-based nanoparticles has attracted remarkable interest from researchers because of their biocompatibility, non-toxicity, low cost and use as disinfectants. SNPs have been extracted and tested from various botanical sources and developed using different preparation methods. SNPs have been used in various applications, such as reinforcement in polymer matrices, Pickering emulsions, antimicrobial agents, encapsulating agents, fat substitutes, etc., and caused increases in specific properties of the resulting product. In the future, to further expand their application field, we can use SNPs to provide new solutions, especially in food products, as a constituent component for the production of more innovative products from organoleptic and utilisation perspectives, such as the development of low-fat products and functional foods. Significant increases in the absorption of bioactive compounds increase their bioavailability and bioactivity. Thus, further research should optimise the production process of SNPs and determine the potential effects of functional products from SNPs.

## Figures and Tables

**Figure 1 polymers-15-01167-f001:**
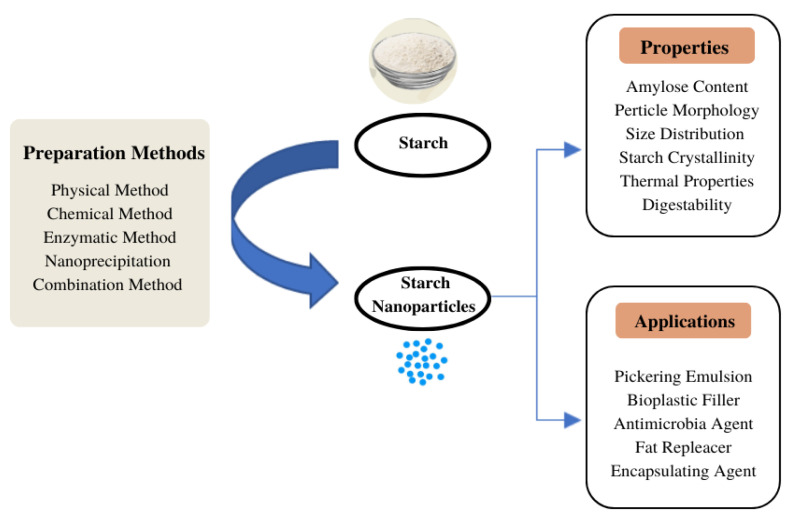
Scheme of SNP preparation, properties and applications.

**Figure 2 polymers-15-01167-f002:**
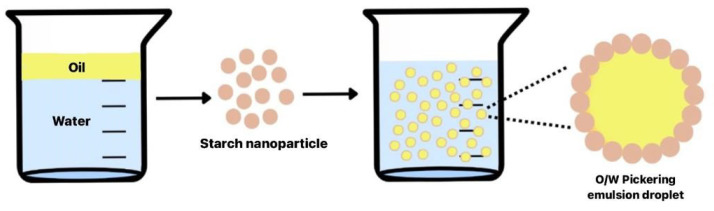
Application of starch nanoparticles as Pickering emulsion.

**Table 1 polymers-15-01167-t001:** Preparation of SNPs by various methods.

Starch Source	Preparation Method	Preparation Condition	%Yield	Ref.
Top-down methods				
Cassava Waxy maize	Gamma irradiation	Doses 20 kGy (14 kG/h)	NR	[46]
Cassava	Gamma irradiation	Doses 20 kGy	NR	[47]
Green Sago	High-pressure homogenization	250 Mpa/5 passes 1 h (Refrigerated for 30 min/after time)	NR	[48]
High amylose maize	High-pressure homogenization	Starch was dispersed high-pressure homogenization was performed at 140, 200, and 250 MPa for 1–4 cycles	NR	[49]
Waxy maize	Ultrasonication	Ultrasonication (80% power, 8 °C, 20 kHz, 75 min	NR	[50]
Waxy maize	Ultrasonication	Sonication (80% power, 8 °C, 24 kHz, 75 min	NR	[51]
Cassava	Ultrasonication	Ultrasonication 8 ◦C, 24 kHz, 75 min	NR	[52]
Corn	Ultrasonication	mixture water-isopropanol (50/50 wt%) ultrasonication (100% power, 10 °C, 20 kHz, 75 min	NR	[53]
Quinoa Maize	Ultrasonication	The suspension is heated in solution NaOH (ultrasonication 20 kHz, 30 min)	NR	[54]
Waxy maize	Acid hydrolysis	3.16 M H_2_SO_4_, hydrolysis at 40 °C for 5 days	NR	[46]
Mung bean	Acid hydrolysis	3.16 M H_2_SO_4_, hydrolysis at 40 °C for 7 days	33.2	[55]
Waxy maize Normal maize High AM maize Potato Mungbean	Acid hydrolysis	3.16 M H_2_SO_4_, hydrolysis at 40 °C for 7 days	NR	[56]
Waxy maize High amylose maize	Acid hydrolysis	3.16 M H_2_SO_4_ hydrolysis at 40 °C for 6 days	NR	[57]
Water chesnut	Acid hydrolysis	3.16 M H_2_SO_4_ hydrolysis at 40 °C for 7 days	27.5	[38]
Andean potato	Acid hydrolysis	3.16 M H_2_SO_4_ hydrolysis at 40 °C for 5 days	NR	[58]
Waxy rice	Acid hydrolysis	2.2 M HCl hydrolysis at 35 °C for 7–10 days	NR	[59]
Sago	Acid hydrolysis	2.2 M HCl hydrolysis 35 °C for 12–48 h	72–80	[39]
Sago	Combined acid hydrolysis and precipitation	HCl 2.2 M hydrolysis 35 °C for 12–48 h, then precipitation with ethanol HCl 2.2 M hydrolysis 35 °C for 12–48 h, then precipitation with butanol	20–25% 22–23%	[39]
Andean potato	Combined acid hydrolysis and –ultrasonication	3.16 M H_2_SO_4_ hydrolysis at 40 °C for 5 days, then sonication 4 °C, 26 kHz	NR	[58]
Waxy maize SNP	Combined cid hydrolysis and ultrasonication	3.16 M H_2_SO_4_ hydrolysis at 4/40 °C for 1–6 days, then ultrasonication 20 kHz, 3 min	78%	[60]
Tapioca	Combined nanoprecipitations and ultrasonication	Precipitation using aceton, then ultrasonication 60 min, 20 kHz, and 150 W	NR	[61]
Lotus seed	Combined enzymatic hydrolysis and ultrasonication	Hydrolysis with pullulanase enzyme (pH 4.6) at (30 ASPU/g of dry starch), 58 °C, 8 h, then ultrasonication 25 ± 1 kHz. Acid Stable Pullulanase Units (ASPU) is ASPU is defined as the amount of enzyme that liberates 1.0 mg glucose from starch in 1 min at pH 4.4 and 60 °C	NR	[62]
Waxy maize	Combined enzymatic hydrolysis and recrystallization	Hydrolysis with pullulanase enzyme (pH 5) at (30 ASPU/g of dry starch), 58 °C, 8 h. Followed by recrystallization at 4 °C 8 h	85%	[63]
Elephant foot yam	Combined enzymatic hydrolysis and recrystallization	Debranching by pullulanase, followed by recrystallization at 4 °C 12–24 h	56.66–61.33%	[64]
Waxy maize	Combined enzymatic hydrolysis and recrystallization	Hydrolysis with pullulanase enzyme at 58 ^O^C 24 h, then recrystallized 5 °C	NR	[40]
b. Top-down methods				
Dry high amylose Corn Potato Tapioca Sweet potato Waxy corn	Nanoprecipitation	Absolute ethanol as a precipitate	NR	[6]
Green banana	Nanoprecipitation	Starch mixed in acetone and precipitated with water	NR	[37]
Tapioca	Nanoprecipitation	Produced with acetone	NR	[61]
Waxy maize	Nanoprecipitation	Starch mixed with ethanol	NR	[40]
Arrowroot	Nanoprecipitation	Produced by butanol	20.65–23.8	[65]
Potato	Nanoprecipitation	Produced by ethanol	NR	[66]

NR = Not reported.

**Table 2 polymers-15-01167-t002:** Morphological characteristics and size of SNPs.

Source	Preparation Method	Shape	Size (nm)	Ref.
Cassava	Gamma irradiation	Agglomerates	50	[47]
Cassava Waxy maize	Gamma irradiation	Laminar laminar Aggregates are formed	20 20–30	[46]
High amylose maize	High-pressure homogenization	Aggregates and porous.	540	[49]
Green sago	High-pressure homogenization	Spherical	23.112	[48]
Cassava	Ultrasonication	Spherical	77.51	[52]
Quinoa Maize	Ultrasonication	flaky and porous flaky and porous	99 214	[54]
Waxy maize	Ultrasonication	Platelet-like	40	[53]
Waxy maize	Ultrasonication	Ellipsoidal	37	[50]
Waxy maize Normal maize High AM maize Potato Mungbean	Acid hydrolysis	Round or oval shapes	41.4 41.0 69.7 43.2 53.7	[56]
Andean potato	Acid hydrolysis	Elliptical-polyhedral shape	132.56–263.38	[58]
Waxy rice	Acid hydrolysis	Round but irregular	220–279.4	[59]
Unripe plantain fruits	Acid hydrolysis	Oval shape but fractured granules.	NR	[83]
Waxy maize High amylose maize	Acid hydrolysis	flat elliptical Round-polygonal	>500 268	[57]
Mungbean	Acid hydrolysis	slightly oval/irregular	141.772	[55]
Water chesnut	Acid hydrolysis	Irregular and rough surface	396	[38]
Sago	Acid hydrolysis	NR	789.30	[39]
Dry high-amylose corn Pea potato Corn Tapioca Sweet potato Waxy corn	Nanoprecipitation	Spherical and elliptical	20–80 30–150 50–225 15–80 30–110 40–100 20–200	[6]
Potato	Nanoprecipitation	Spherical and elliptical	50–150	[66]
Arrowroot	Nanoprecipitation	non-granular morphologies with porous	261.4	[65]
Green banana	Nanoprecipitation	NR	135.1	[37]
Waxy maize starch	Nanoprecipitation	Irregular	201.67	[40]
Tapioca	Nanoprecipitation	Spherical	219	[61]
Lotus seed	Enzymatic hydrolysis	irregular shapes	NR	[62]
Waxy rice	Enzymatic hydrolysis	Irregular shape	500	[74]
Waxy maize	Enzymatic hydrolysis	Irregular with erosion surface	NR	[75]
Sago	Combined acid hydrolysis and precipitation method with butanol Combined acid hydrolysis and precipitation method with ethanol	NR NR	7.57–178 21.98–97.50	[39]
Andean potato	Combined acid hydrolysis and ultrasonication	elliptical-polyhedral shape	153.63–366.76	[58]
Potato	Combined acid hydrolysis and ultrasonication	Spherical	40	[84]
Waxy maize	Combined acid hydrolysis and ultrasonication	Globular	40–90	[60]
Tapioca	Combined nanoprecipitation and ultrasonication	Spherical	163	[61]
lotus seed	Combined enzyme hydrolysis and ultrasonication	irregular shapes with the uneven surface	16.7–2420	[62]
Elephant foot yam	Combination enzyme and recrystallization	irregular to spherical shapes	182.07–198.1	[64]
Waxy maize	Combined enzyme hydrolysis and recrystallization	Spherical microscale coralloid aggregates	156	[40]
Waxy maize	Combined enzyme hydrolysis and recrystallization	Irregular	80–120	[63]

NR: Not Reported.

**Table 3 polymers-15-01167-t003:** Crystallinity of starch nanoparticles from various preparation methods.

Starch Source	Preparation Method	Crystallinity (%)	Crystalline Type		Ref.
			Native Starch	NSPs	
Cassava Waxy maize	Gamma irradiation	Decrease Decrease	NR NR	Amorphous Amorphous	[46]
High amylose maize starch	High-pressure Homogenization	7.8	B-type	B-type	[49]
Cassava	Ultrasonication	Decrease	C-Type	Amorphous	[52]
Quinoa Maize	Ultrasonication	Decrease Decrease	A-Type A-Type	Amorphous Amorphous	[54]
Waxy maize	Ultrasonication	-	A-Type	Amorphous	[53]
Waxy maize	Ultrasonication	Decrease	A-Type	Amorphous	[50]
High AM maize Potato	Acid hydrolysis	61.4 89.4	A-type B-type	B-type B-type	[56]
Andean potato	Acid hydrolysis	42.2	B-type	B-type	[58]
Waxy rice	Acid hydrolysis	No change	A-type	A-type	[59]
Waxy maize High amylose maize	Acid hydrolysis	NR NR	A-type B-type	A-type A-type	[57]
Waxy maize	Acid hydrolysis	53	NR	A-type	[46]
Sago	Acid hydrolysis	36	NR	NR	[39]
Dry high amylose corn Pea Potato Corn Tapioca Sweet potato Waxy corn	Nanoprecipitation	39.8 31.5 26.3 23.2 19.3 20.7 7.1	B-type C-type B-type A-type A-type A-type A-type	V-type	[6]
Potato	Nanoprecipitation	23.5	B-type	V-type	[66]
Arrowroot	Nanoprecipitation	28.36–45.12	A-type	V-type	[65]
Waxy maize	Nanoprecipitation	NR	NR	V-type	[40]
Tapioca	Nanoprecipitation	12.53	A-type	V-type	[61]
lotus seed	Enzyme hydrolysis	65.07	B-type	B-type	[62]
Waxy rice	Enzymatic hydrolysis	NR	A-type	A-type	[74]
Waxy maize	Enzymatic hydrolysis	Increase	A-type	NR	[75]
Sago	Combined acid hydrolysis and precipitation method with ethanol Combined acid hydrolysis and precipitation method with butanol	41 34	NR NR	NR NR	[39]
Waxy maize	Combined cid hydrolysis andultrasonication	27.68	A-type	A-type	[60]
Tapioca	Combined nanoprecipitation and ultrasonication	6.49–15.21	A-type	V-type	[61]
lotus seed	Combined enzyme hydrolysis and ultrasonication	57.5–61.3	B-type	B-type	[62]
Elephant foot yam	Combined enzyme and recrystallization	41.30–43.22	C-type	B-type	[64]
Waxy maize starch	Combined enzyme hydrolysis and recrystallization	NR	NR	B +V-type	[40]
Waxy maize starch	Combined enzyme hydrolysis and recrystallization	45.28	A-type	B +V-type	[63]

NR: Not Reported.

**Table 4 polymers-15-01167-t004:** Thermal characteristics of starch nanoparticles.

Starch Source	Preparation Method	Technique	Result	Ref.
Cassava, waxy maize starch	Gamma irradiation	TGA	Degraded at a lower temperature than native starch and sudden decrease in weight loss	[46]
Green sago	High-Pressure homogenization	TGA	High degradation temperature	[48]
Cassava	Ultrasonication	TGA/DSC	SNPs are more thermally unstable and have low gelatinization temperature	[52]
Quinoa Maize	Ultrasonication	DSC	T_o_, T_p_, T_c_ and ∆H decreased T_o_, T_p_ and T_c_ decreased but ∆H increased	[54]
Waxy maize	Ultrasonication	DSC	∆H decreased	[53]
Waxy maize	Ultrasonication	DSC	∆H decreased	[50]
Waxy maize Normal maize	Acid hydrolysis	DSC	T_p_ and T_c_ and ∆H, but T_o_ decreased	[56]
Waxy rice	Acid hydrolysis	DSC	T_p_ and T_c_ increased as the hydrolysis time increased, but T_o_ and ΔT decreased	[59]
Unripe plantain fruit	Acid hydrolysis	DSC	T_p_, T_c_ and ΔT increased as the hydrolysis time increased, but T_o_ decreased	[83]
Potato	Nanoprecipitation	TGA	Thermal degradation of SNPs started earlier than for native starch	[66]
Arrowroot	Nanoprecipitation	DSC	T_p_ decreased, T_o_ and ∆H increased	[65]
High amylose corn	Nanoprecipitation	DSC	T_o_, T_p_ and T_c_ decreased, but ∆H increased	[6]
Potato	Nanoprecipitation	DSC	T_o_ and ∆H decreased, but T_p_ increased	
Pea corn Tapioca Sweet potato Waxy corn	Nanoprecipitation	DSC	T_o_, T_p_, T_c_ and ∆H decreased	
Tapioca	Combined nanoprecipitation and ultrasonication	DSC	T_o_, T_p_, T_c_ and ∆H decreased	[61]
Elephant foot yam	Combined enzyme and recrystallization	DSC	T_p_, T_o_, T_c_ increased, but T_c_ decreased at 24 h of hydrolysis	[64]
Waxy maize starch	Combined enzyme hydrolysis and recrystallization	TGA	Maximum degradation temperature decreased	[63]

**Table 5 polymers-15-01167-t005:** Application of SNPs compared with SNCs as Pickering emulsion.

Type Starch	Aqueous Phase	Oil Phase	Emulsion Type	Emulsification Method	Result	Ref.
Maize SNC	Water	Paraffin	o/w	Homogenization (10,000 rpm, 4 min)	- The emulsion is very stable up to 2 months of storage - Creaming is wholly inhibited at 6% SNC concentration	[114]
Taro SNP	NaCl	MCT oil	o/w	Homogenization (12,000 rpm, 2 min)	Emulsion with the best stability at an SNP concentration of 7% with an oil fraction of 0.5, up to 28 days	[115]
OSA amaranth and maize SNC	Phosphate buffer pH 7 (NaCl 0.2 M)	Canola oil	o/w	High shear mixer (22,000 rpm, 3 min)	the best emulsion stability on amaranth OSA starch nanocrystals (emulsion index 1.0 ± 0.02), for 10 days of storage	[116]
Waxy Maize SNC	Water	Parrafin	o/w and w/o	pH difference using HCl or NaOH, homogenization (10,000 rpm, 4 min)	- Decreased droplet rate and creaming as SNC increase - pH has no significant effect on creaming ability, but the emulsion stability significantly decreases at low pH	[117]
Tapioca, corn, and sweet potato SNC	NaCl	Soybean oil	o/w	High-speed homogenizer (10,000 rpm, 2 min)	- Creaming index 18–22% with emulsion drop size 29–32 m after 1 year of storage - Medium particle size produces the best emulsion stability (100–220 nm) - Best emulsion stability on corn starch suspension (2%) and oil fraction (0.5)	[34]
Breadfruit SNC	NaOH (0.1875 and 0.375 M)	MCT oil	o/w	Homogenization (10,000 rpm, 5 min)	Treatment of 5% starch concentration with 0.1875 M NaOH resulted in the best starch stability for 2 weeks of storage with the lowest cream index and the smallest droplets.	[118]
Maize SNC	Water	Corn oil	o/w	Homogenization (20,000 rpm, 3 min)	- Addition of SNP >9.1% can increase emulsion stability up to 95% - Emulsions using SNPs with a diameter of <30 nm produced the best increase in instability	[99]
Waxy maize SNC	Water	MCT oil	o/w	High-speed homogenizer (18,000 rpm, 4 min)	No o/w emulsion phase separation was detected during 30 days of storage	[119]
Corn SNC	Water	Sunflower oil	o/w	Homogenization (12,000 rpm, 5 min)	No cream was observed in the emulsion after storage for 6 months.	[120]
Oxidation of cassava, corn, and bean SNC	Water	Soybean oil	o/w	Homogenization 1 min	- Oxidated nanocrystals produce a stable suspension for up to 21 days - The addition of nanocrystals of cassava starch produces the best emulsion	[121]

o/w: oil in water, w/o: water in oil.

**Table 6 polymers-15-01167-t006:** Application of SNPs compared with SNCs as Fillers in Bioplastics.

Manufacturing Technique	Bioplastic Composition	Result	Ref.
Casting	Pea starch (5 g) + glycerol (1.5 g) + waxy maize acid hydrolysis SNC (5%)	There was a decrease in elongation, YM and TS by 57%, 305%, 73%, respectively. WVP reduction up to 62%	[133]
Casting	PU resin (polyurethane) + maize corn acid hydrolysis-ultrasonication SNP (20%)	The addition of 20% SNPs reduces WVP to 60% and oxygen permeability decreases to 75%. There was an increase in the value of T_g_, T_m_, and ∆H	[134]
Casting	Corn starch (7.5 g + glycerol (3 g) + taro enzymolysis SNP (10%)	Decrease in elongation and WVP by 24% and 56%, respectively, an increase in TS 161%. There was an increase in the value of T_g_, T_m_, and ∆H	[135]
Casting	Amanduble starch + glycerol + amadumbe acid hydrolysis SNC (2.5%)	There was an increase in TS of 62% and a decrease in WVP of 8.7%. There was an increase in the value of T_g_, T_m_, and ∆H	[136]
Casting	Potato starch + glycerol + amadumbe acid hydrolysis SNC (2.5%)	There was an increase in TS 288% and a decrease in WVP 11%	[136]
Casting	Polycaprolactone (PCL) + corn SNP (5%)	There was a decrease in the elongation value of 9%, an increase in YM 12% and TS 44%. T_m_ and ∆H decreased	[137]
Casting	Waterborne polyurethane (WPU) + pea SNC (10%)	The addition of SNCs by 10% showed a decrease in elongation by 27%, but there is an increase in TS and YM by 169% and 3733%, respectively	[138]
Casting	Cross-linked cassava starch + glycerol 2.5 g) + cassava SNC (6%)	Increase in young modulus and tensile strength, but decrease in elongation and water vapor permeability	[139]
Casting	Composite sago starch + sago SNP (6%)	Increase in elongation, YM and TS by 34%, 9% and 8%, respectively. WVP decrease up to 51%	[48]
Extrusion	PBA/TPS (70:30) + glycerol (7.5%), citric acid (0.6%), and stearic acid (0.3%) + cassava gamma irradiation SNP (0.6%)	There was an increase of about 20% in YM and TS. T_g_: −34 °C and T_m_: 117 °C	[47]
Extrusion	PBA/TPS (70:30) + glycerol (7.0%), citric acid (%) and stearic acid (0.3%) + cassava ultrasonication SNP 1%	The addition of 1% SNP can increase elongation by 35%, YM by 36%, TS by 27%, and decrease WVP up to 21.3%	[52]

YM: young modulus; TS: tensile strength; WVP: water vapor permeability; PBAT: polybutylen adipate-co-terephthalate; TPS: thermoplastic starch.

**Table 7 polymers-15-01167-t007:** Application of native and modified SNPs as encapsulation agents.

Type of Starch	Preparation Method	Encapsulation Compound	%Encaptulation Efficiency	Ref.
Native starch				
Banana starch	Nanoprecipitation	Curcumin	85.23	[37]
Waxy maize	Nanoprecipitation	Polyphenols	60–70	[163]
Quiona	Nanoprecipitation	Piroxicam	84	[164]
Insoluble porous starch	Nanoprecipitation	Paclitaxel	73.92	[165]
Soluble SNPs	Ethanol precipitate	Vitamin E	91.63	[166]
Horse chestnut Water chesnut Lotus Stem	Acid hydrolysis	Catechin	59.09 48.30 55	[167]
Quiona Maize starch	Ultasonication	Rutin	67.4 63.1	[54]
Normal corn high-amylose Waxy corn	Ultrasonication	Anthocyanin	52.5 45.5 49.4	[168]
Horse chestnut Lotus Stem Water chesnut	Ball milling	Resveratrol	81.46 75.83 73.37	[169]
b. Modified starch				
Acetylated Banana	Nanoprecipitation	Curcumin	82.23–92.12	[150]
Acetylated Banana	Nanoprecipitation	Curcumin	90.63	[37]
Acetylated corn	Nanoprecipitation	Ciprofloxacin	20.5–89.1	[170]
OSA Waxy maize	Emulsion-diffusion	Conjugated linoleic acid	>97	[171]
Debranched Waxy corn	Enzyme hydrolysis with pullulanase	Epigallocatechin gallate	84.4	[172]
Debranched waxy maize SNPs	Ultrasonication combined with recrystallization	Epigallocatechin gallate	>80	[155]

## Data Availability

Not applicable.

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
