# Peer review of "Starch Nanoparticles: Preparation, Properties and Applications"

_polymers, 2023, doi:10.3390/polym15051167_

Round 1

Reviewer 1 Report

1. This current manuscript needs scientific proofreading. Some sentences are not properly structured, wrong syntax and grammar.
2. Abstract: A summary of the prospects of this research field should be added.
3. "starch nanoparticles (SNPs)" appeared several times in the paper. Since abbreviation was defined, use it uniformly after the first mention.
4. Introduction: It is necessary to add descriptions of starch structure and the general introduction of nanostarch, such as starch nanoparticles and starch nanocrystals, definitions, etc. Or it can condense the "2. Starch Nanoparticle" and incorporate it into the Introduction.
5. Line 62-64: In my understanding, Nanostarch is prepared mainly as starch nanoparticles (SNPs) and starch nanocrystals (SNCs). Is the definition of "starch-based nanoparticles" universally accepted? Please prove it.
6. “Starch Nanoparticle” in current manuscript included “starch nanocrystals” and “starch-based nanoparticles”, therefore, when summarizing the preparation method, it is necessary to indicate what kind of starch nanoparticles of the method is used to prepare, as well as the size and properties of the granules.
7. Some tables have duplicate contents and can be merged.
8. It is necessary to further summarize the digestion rate of different types of nanostarch.
9. The paper should deeply analyze the problems existing in the application of nanostarch in different fields.
10. Please re-check the citation and write-up of references. Some references are old and inaccurate.

Author Response

Thank you very much for your comments concerning our manuscript entitled “Starch Nanoparticles: Preparation, Properties, and Applications”. Those comments are valuable and very helpful for revising and improving our paper. We have studied the comments carefully and have made a correction which we hope meets with approval. The revised portions are marked in red on the paper. The main correction and the responses to the reviewer’s comment are as follows:

  1. This current manuscript needs scientific proofreading. Some sentences are not properly structured, wrong syntax and grammar.

Response:

Thank you for your suggestions. The manuscript has been proofread (the certificate was attached)

  1. Abstract: A summary of the prospects of this research field should be added.

Response:

It has been added (lines 14-20)

  1. "starch nanoparticles (SNPs)" appeared several times in the paper. Since abbreviation was defined, use it uniformly after the first mention.

Response:

Thank you for your suggestion. It has been revised

  1. Introduction: It is necessary to add descriptions of starch structure and the general introduction of nanostarch, such as starch nanoparticles and starch nanocrystals, definitions, etc. Or it can condense the "2. Starch Nanoparticle" and incorporate it into the Introduction.

Response:

It has been added (lines 31-35; 53-56)

  1. Line 62-64: In my understanding, Nanostarch is prepared mainly as starch nanoparticles (SNPs) and starch nanocrystals (SNCs). Is the definition of "starch-based nanoparticles" universally accepted? Please prove it.

Response:

Thank you for your correction. Nanostarch are of two types: starch nanocrystals (SNCs) are crystalline portions resulting from the disruption of amorphous domains in starch granules and starch nanoparticles (SNPs) generated from gelatinised starch may include amorphous regions. This review focused to study nanostarch pacticles (SNPs). It has been revised (lines 53-56)

  1. “Starch Nanoparticle” in current manuscript included “starch nanocrystals” and “starch-based nanoparticles”, therefore, when summarizing the preparation method, it is necessary to indicate what kind of starch nanoparticles of the method is used to prepare, as figure as the size and properties of the granules.

Response:

It has been revised. This review focused to study NSPs and discussion about SNCs was deleted.

  1. Some tables have duplicate contents and can be merged.

Response:

Table 1-4 has been merged (see Table 1)

  1. It is necessary to further summarize the digestion rate of different types of nanostarch.

Response:

It has been added (lines 429-454)

  1. The paper should deeply analyze the problems existing in the application of nanostarch in different fields.

Response:

It has been added (lines 457-463; 529-535)

  1. Please re-check the citation and write-up of references. Some references are old and inaccurate.

Response:

It has been re-checked (see references)

Reviewer 2 Report

The authors need to highlight in the abstract the importance of starch Nps and the most important applications.

The authors must analyze the information and not only present the tables; there must be own comments from the authors to achieve a contribution to the review is necessary to complete this part of the document.

Author Response

Thank you very much for your comments concerning our manuscript entitled “Starch Nanoparticles: Preparation, Properties, and Applications”. Those comments are valuable and very helpful for revising and improving our paper. We have studied the comments carefully and have made a correction which we hope meets with approval. The revised portions are marked in red on the paper. The main correction and the responses to the reviewer’s comment are as follows:

The authors need to highlight in the abstract the importance of starch Nps and the most important applications.

Response:

It has been added (see abstract lines 10-20)

The authors must analyze the information and not only present the tables; there must be own comments from the authors to achieve a contribution to the review is necessary to complete this part of the document.

Response:

It has been added (lines 113-114, 130-133, 166-168, 200-202, 221-263)

Reviewer 3 Report

The main problem with this otherwise interesting review article is the numerous mistakes in english language. The manuscript would greatly benefit from the careful and thorough reading of an english-speaking person in order to prevent many ambiguities. The manuscript is very long and it is not the duty of a reviewer to extensively correct english. I listed below some mistakes I spotted but there are many other that must be corrected before acceptance for publication in Polymers. I also indicated comments and aspects that I would like to be addressed by the authors.

·       Title: I would use "nanoparticles" (plural) since, as demonstrated by this review article, there are different types of starch particles.

·       Lines 10-13: please rephrase for english. In addition, there is an ambiguity in the section in lines 10-11. There are two ways to produce starch particles, as explained in lines 73-80. It is not clear here what "modification of starch molecules" mean. The better distinction made at the beginning of section 2 should be used.

·       Line 19: delete "recorded".

·       Line 25: Starch **is** a carbohydrate. Starch **granules** are a source of carbohydrates.

·       Line 31: rephrase for english. Starch is constituted of two components, namely amylose…

·       Lines 39-42: be precise about the limitations.

·       Line 48: "SNPs" have been defined here. Please use this abbreviation for the rest of sections 1 and 2, and not "starch nanoparticles (SNPs)". That's the purpose of an abbreviation.

·       Line 54: Pickering emulsion stabilizers

·       Line 64: …which results from a precipitation/ recrystallization from a starch solution.

·       Line 68: What does "not exceeding the size of one molecule" mean?

·       Line 70: delete "entirely".

·       Line 75: "refinement" is not the proper word. "fragmentation" perhaps?

·       Line 90: delete "extraction".

·       Line 104 and legend Table 1: physical methods (plural)

·       Line 106: these methods

·       Lines 125-126: please rephrase for english

·       Line 129: what does "pins use 24 kHz power" mean?? If the reader directly applies a 75 min treatment, he will rapidly obtain a gelatinized goo, only because of significant hydrothermal effect and cavitation-enhanced degradation. Please be more precise.

·       Line 132: "does **not" use…", perhaps?

·       Section 3.2 (Chemical methods – plural!): there is some confusion here. Acid hydrolysis is a chemical method since it uses a chemical reaction. Nanoprecipitation is not since what is mentioned here is merely a precipitation/crystallization process from solution using a precipitant/complexing agent (this is what is described in lines 161-169 although this section is very poor in references). The two approaches must be separated, although they can be combined, as described in section 3.4. Acid hydrolysis is a top-down fragmentation of starch granules while nanoprecipitation is a bottom-up one.

·       In addition, the methods described here will have a different effect on amylose and amylopectin solutions. Using the alcohols mentioned here will crystallize *amylose* by forming a complex. Amylopectin does not form complexes with these alcohols. There is a very rich literature on forming complexes, hence particles. Ref. 23 is not enough to illustrate this method. Line 143: what is described here is the recrystallization of the amylose

·       Line 149: please rephrase for english.

·       In this section, it should be reminded that acid-hydrolysis of starch granules is a rather old method (see Lintner's and Naegeli's methods) but the authors only mentioned a few articles that are very recent. This should be improved. The authors can refer to the introduction of this paper (DOI:10.1021/bm0340422) which contains several references, which, by the way, provide a clear illustration of what nanocrystals from acid-hydrolyzed waxy maize starch look like, or this other review article (DOI:10.1080/10408398.2012.684551).

·       Line 224: the amylose content may vary depending on the botanical source of the starch granules.

·       Line 232: hydrolysis

·       Line 233: please rephrase this sentence.

·       In addition, in this section: of course, the amylose content will be increased by precipitation with an alcohol since only amylose forms complexes with these alcohols. 

·       Line 247: use "Particle morphology and size distribution" to avoid confusion with starch granules.

·       Line 248: The morphology, structure, and size distribution of SNPs can be characterized…

·       Line 285: Again, nanoprecipitation is not a chemical method to prepare SNPs.

·       Line 314: Kumari et al did not invent this definition which is the general definition of the crystallinity degree of any sample. If you need a reference for structural aspects, please use a more relevant review article like DOI: 10.1002/star.201000013. Please also explain for the non-specialist reader what are A, B, C and V-type structures (DOI: 10.1071/CH07168).

·       Table 5: it might be useful to have a column describing the crystal type of the native starch granules (before treatment), since the processing may result in a crystallinity change.

·       Line 355: please rephrase for english.

·       Line 392: what does "interference" mean? Rephrase.

·       Line 452: Pickering is the name of a scientist and should be capitalized. Chen et al. did not invent the de definition of a Pickering emulsion. Please use a better reference or a review article on the subject.

·       Line 475: to stabilize Pickering emulsions.

·       In Table 7: please correct several "Homogenisasi"

·       Table 9: strictly, acetylated or succinylated starch are not starch any longer, hence the totally different properties, in particular in specific formulations. So I would separate unmodified and chemically-modified starch samples into two tables. Please define OSA.

·       References: please use a homogenized style for journal names (full or abbreviated).

·       Ref. 34: Dufresne

Author Response

Thank you very much for your comments concerning our manuscript entitled “Starch Nanoparticles: Preparation, Properties, and Applications”. Those comments are valuable and very helpful for revising and improving our paper. We have studied the comments carefully and have made a correction which we hope meets with approval. The revised portions are marked in red on the paper. The main correction and the responses to the reviewer’s comment are as follows:

Reviewer 1

  1. This current manuscript needs scientific proofreading. Some sentences are not properly structured, wrong syntax and grammar.

Response:

Thank you for your suggestions. The manuscript has been proofread (the certificate was attached)

  1. Abstract: A summary of the prospects of this research field should be added.

Response:

It has been added (lines 14-20)

  1. "starch nanoparticles (SNPs)" appeared several times in the paper. Since abbreviation was defined, use it uniformly after the first mention.

Response:

Thank you for your suggestion. It has been revised

  1. Introduction: It is necessary to add descriptions of starch structure and the general introduction of nanostarch, such as starch nanoparticles and starch nanocrystals, definitions, etc. Or it can condense the "2. Starch Nanoparticle" and incorporate it into the Introduction.

Response:

It has been added (lines 31-35; 53-56)

  1. Line 62-64: In my understanding, Nanostarch is prepared mainly as starch nanoparticles (SNPs) and starch nanocrystals (SNCs). Is the definition of "starch-based nanoparticles" universally accepted? Please prove it.

Response:

Thank you for your correction. Nanostarch are of two types: starch nanocrystals (SNCs) are crystalline portions resulting from the disruption of amorphous domains in starch granules and starch nanoparticles (SNPs) generated from gelatinised starch may include amorphous regions. This review focused to study nanostarch pacticles (SNPs). It has been revised (lines 53-56)

  1. “Starch Nanoparticle” in current manuscript included “starch nanocrystals” and “starch-based nanoparticles”, therefore, when summarizing the preparation method, it is necessary to indicate what kind of starch nanoparticles of the method is used to prepare, as figure as the size and properties of the granules.

Response:

It has been revised. This review focused to study NSPs and discussion about SNCs was deleted.

  1. Some tables have duplicate contents and can be merged.

Response:

Table 1-4 has been merged (see Table 1)

  1. It is necessary to further summarize the digestion rate of different types of nanostarch.

Response:

It has been added (lines 429-454)

  1. The paper should deeply analyze the problems existing in the application of nanostarch in different fields.

Response:

It has been added (lines 457-463; 529-535)

  1. Please re-check the citation and write-up of references. Some references are old and inaccurate.

Response:

It has been re-checked (see references)

Thank you very much for your comments concerning our manuscript entitled “Starch Nanoparticles: Preparation, Properties, and Applications”. Those comments are valuable and very helpful for revising and improving our paper. We have studied the comments carefully and have made a correction which we hope meets with approval. The revised portions are marked in red on the paper. The main correction and the responses to the reviewer’s comment are as follows:

The authors need to highlight in the abstract the importance of starch Nps and the most important applications.

Response:

It has been added (see abstract lines 10-20)

The authors must analyze the information and not only present the tables; there must be own comments from the authors to achieve a contribution to the review is necessary to complete this part of the document.

Response:

It has been added (lines 113-114, 130-133, 166-168, 200-202, 221-263)

Thank you very much for your comments concerning our manuscript entitled “Starch Nanoparticles: Preparation, Properties, and Applications”. Those comments are valuable and very helpful for revising and improving our paper. We have studied the comments carefully and have made a correction which we hope meets with approval. The revised portions are marked in red on the paper. The main correction and the responses to the reviewer’s comment are as follows:

The main problem with this otherwise interesting review article is the numerous mistakes in english language. The manuscript would greatly benefit from the careful and thorough reading of an english-speaking person in order to prevent many ambiguities. The manuscript is very long and it is not the duty of a reviewer to extensively correct english. I listed below some mistakes I spotted but there are many other that must be corrected before acceptance for publication in Polymers. I also indicated comments and aspects that I would like to be addressed by the authors.

Response:

Thank you for your suggestions. The manuscript has been proofread (the certificate was attached)

  1. Title: I would use "nanoparticles" (plural) since, as demonstrated by this review article, there are different types of starch particles.

Response:

It has been revised (Line 2)

  1. Lines 10-13: please rephrase for english. In addition, there is an ambiguity in the section in lines 10-11. There are two ways to produce starch particles, as explained in lines 73-80. It is not clear here what "modification of starch molecules" mean. The better distinction made at the beginning of section 2 should be used.

Response:

It has been revised (Line 14-20)

  1. Line 19: delete "recorded".

Response:

It has been deleted (Line 19)

  1. Line 25: Starch **is** a carbohydrate. Starch **granules** are a source of carbohydrates.

Response:

It has been revised (Line 25)

  1. Line 31: rephrase for english. Starch is constituted of two components, namely amylose…

Response:

It has been revised (Lines 31-35)

  1. Lines 39-42: be precise about the limitations.

Response:

It has been added (Lines 42-43)

  1. Line 48: "SNPs" have been defined here. Please use this abbreviation for the rest of sections 1 and 2, and not "starch nanoparticles (SNPs)". That's the purpose of an abbreviation.

Response:

It has been revised

  1. Line 54: Pickering emulsion stabilizers

Response:

It has been revised (Line 59)

  1. Line 64: …which results from a precipitation/ recrystallization from a starch solution.

Response:

This sentence was deleted

  1. Line 68: What does "not exceeding the size of one molecule" mean?

Response:

It has been revised (Line 72)

  1. Line 70: delete "entirely".

Response:

It has been revised (Line 74)

  1. Line 75: "refinement" is not the proper word. "fragmentation" perhaps?

Response:

It has been revised (Line 80)

  1. Line 90: delete "extraction".

Response:

It has been revised (Line 93)

  1. Line 104 and legend Table 1: physical methods (plural)

Response:

It has been revised (Line 106)

  1. Line 106: these methods

Response:

It has been revised (Line 111)

  1. Lines 125-126: please rephrase for English

Response:

It has been revised (Lines 125-126)

  1. Line 129: what does "pins use 24 kHz power" mean?? If the reader directly applies a 75 min treatment, he will rapidly obtain a gelatinized goo, only because of significant hydrothermal effect and cavitation-enhanced degradation. Please be more precise.

Response:

It has been revised (Lines 128-130)

  1. Line 132: "does **not" use…", perhaps?

Response:

It has been revised (Line 135)

  1. Section 3.2 (Chemical methods – plural!): there is some confusion here. Acid hydrolysis is a chemical method since it uses a chemical reaction. Nanoprecipitation is not since what is mentioned here is merely a precipitation/crystallization process from solution using a precipitant/complexing agent (this is what is described in lines 161-169 although this section is very poor in references). The two approaches must be separated, although they can be combined, as described in section 3.4. Acid hydrolysis is a top-down fragmentation of starch granules while nanoprecipitation is a bottom-up one.

Response:

It has been revised (Line 137). Thank you for your correction. Nanoprecipitation is not a chemical method. It has been separated from this section.

  1. In addition, the methods described here will have a different effect on amylose and amylopectin solutions. Using the alcohols mentioned here will crystallize *amylose* by forming a complex. Amylopectin does not form complexes with these alcohols. There is a very rich literature on forming complexes, hence particles. Ref. 23 is not enough to illustrate this method. Line 143: what is described here is the recrystallization of the amylose

Response:

It has been added (Lines 173-192)

  1. Line 149: please rephrase for english.

Response:

It has been revised (Lines 146-149)

  1. In this section, it should be reminded that acid-hydrolysis of starch granules is a rather old method (see Lintner's and Naegeli's methods) but the authors only mentioned a few articles that are very recent. This should be improved. The authors can refer to the introduction of this paper (DOI:10.1021/bm0340422) which contains several references, which, by the way, provide a clear illustration of what nanocrystals from acid-hydrolyzed waxy maize starch look like, or this other review article (DOI:10.1080/10408398.2012.684551).

Response:

It has been revised (Lines 139-145)

  1. Line 224: the amylose content may vary depending on the botanical source of the starch granules.

Response:

It has been revised (Lines 229-230)

  1. Line 232: hydrolysis

Response:

It has been revised (Lines 236-237)

  1. Line 233: please rephrase this sentence.

Response:

It has been revised (Lines 238-240)

  1. In addition, in this section: of course, the amylose content will be increased by precipitation with an alcohol since only amylose forms complexes with these alcohols. 

Response:

It has been revised (Lines 238-240)

  1. Line 247: use "Particle morphology and size distribution" to avoid confusion with starch granules.

Response:

It has been revised (Line 251)

  1. Line 248: The morphology, structure, and size distribution of SNPs can be characterized…

Response:

It has been revised (Lines 252-255)

  1. Line 285: Again, nanoprecipitation is not a chemical method to prepare SNPs.

Response:

It has been deleted

  1. Line 314: Kumari et al did not invent this definition which is the general definition of the crystallinity degree of any sample. If you need a reference for structural aspects, please use a more relevant review article like DOI: 10.1002/star.201000013. Please also explain for the non-specialist reader what are A, B, C and V-type structures (DOI: 10.1071/CH07168).

Response:

It has been revised (Lines 314-326)

  1. Table 5: it might be useful to have a column describing the crystal type of the native starch granules (before treatment), since the processing may result in a crystallinity change.

Response:

It has been added (Table 5)

  1. Line 355: please rephrase for english.

Response:

It has been revised (Line 358)

  1. Line 392: what does "interference" mean? Rephrase.

Response:

It has been revised (Lines 391-393)

  1. Line 452: Pickering is the name of a scientist and should be capitalized. Chen et al. did not invent the de definition of a Pickering emulsion. Please use a better reference or a review article on the subject.

Response:

It has been revised (Lines 463-465)

  1. Line 475: to stabilize Pickering emulsions.

Response:

It has been revised (Lines 486-487)

  1. In Table 7: please correct several "Homogenisasi"

Response:

It has been revised (Table 5)

  1. Table 9: strictly, acetylated or succinylated starch are not starch any longer, hence the totally different properties, in particular in specific formulations. So I would separate unmodified and chemically-modified starch samples into two tables. Please define OSA.

Response:

It still in one table but it has been separated in point a and b (Table 7)

  1. References: please use a homogenized style for journal names (full or abbreviated).

Response:

It has been checked and revised (see references)

  1. 34: Dufresne

Response:

It has been revised (Ref no.29)

Round 2

Reviewer 2 Report

No comment

Author Response

Thank you

Reviewer 3 Report

The authors have considered most comments and corrections of the reviewers and have revised their manuscript accordingly. In particular, they mentioned that the text has been proofread by an English-speaking person. It is only partially true and a significant effort must still be done to improve the language and prevent ambiguities before publication. Some corrections and questions are indicated below and should be addressed.

·       Line 32: Starch is a mixture of two macromolecules, namely…

·       Line 52: nanostarches

·       Lines 53-54: this sentence is ambiguous. Please rephrase for english.

·       Line 57: and which may include…

·       Line 62: This review focuses on SNPs, their various…

·       Line 65: delete the first "of SNPs"

·       Line 66: The scheme in Figure 1 summarizes the preparation, properties and applications of SNPs.

·       Line 69: Figure 1: Scheme of SNP preparation, properties and applications

·       Line 79: On the one hand, top-down methods…

·       Line 81: of large particles into small ones

·       Line 82: On the other hand, bottom-up methods…

·       Line 85: to produce nanoparticles.

·       Line 89: I am not sure of what "sophisticated" (which is a personal qualification with which some readers might object) mean here. Equipment to fragment starch granules can also be called "sophisticated". "specific" maybe? But both approaches use specific equipment. I would rephrase anyway.

·       Line 101: Preparation Methods of Starch nanoparticles (all plural)

·       Table 1: why don't you make a clear separation in the table between Top-down and bottom-up methods as described in the text?

·       Table 1: be careful that most numbers contain commas! Please correct. Use the same sign for "°C". It varies too much. Define ASPU.

·       Line 114: Besides, physical methods…

·       Line 114: perhaps they are less time-consuming but are they less energy-consuming (a crucial factor in the process design and industrial scale-up)?

·       Line 117: the breakdown of macromolecules into smaller molecules is simply called "depolymerization".

·       Line 128: The sentence "Ultrasonication destroys microwaves" does not mean anything. Please describe ultrasonication mechanism more accurately (especially because it is not related to microwave irradiation!).

·       Line 136: The yield is higher because acid hydrolysis partly dissolves away the material.

·       Line 141 which was applied by Naegeli…….. whereas Lintner used hydrochloric acid…

·       Line 144: starch granules

·       Line 144: delete "using hydrochloric or sulfuric acid" since it has been written above.

·       Line 148: No need to repeat for the third time what was written above. The reader understood that sulfuric or hydrochloric acids were used for the hydrolysis. 

·       Line 151: "SNPs" instead of "starch"

·       Line 157: "sulfate groups" (according to IUPAC nomenclature)

·       Enzymatic methods: perhas t would be helpful to explain the specific action of each enzyme and why they are used to degrade starch granules.

·       Line 167: This is the enzymatic section. Why do you talk about acid hydrolysis here?

·       Line 171: "combined methods" or "combination of methods"

·       Line 178: not "polymer"; "nanoscale particle"

·       Line 201 and rest of section 3.5: Combined methods

·       Line 228: Starch Nanoparticles

·       Section 4.1: Why is only amylose content only considered and why is it particularly important? Why not amylopectin, which also influences the properties? The overall *composition*, hence the amylose/amylopectin ratio, is important. 

·       Line 257: please rephrase "the SNP scanning technique" that does not mean anything.

·       Lines 263-264: "SNPs can be…..   The differences in morphology depend….

·       Line 266: "particles", not "molecules"

·       Line 299: SNPs are spherical…..

·       Line 320: it is not exclusively ascribed. It is *mostly* ascribed. Amylose can participate to crystallites to some extent.

·       Ref. 84 is not relevant to explain the differences in the starch crystal structure. Please use proper review articles or textbook chapters, such as Perez et al. (https://doi.org/10.1002/star.201000013) or Buléon et al. (https://doi.org/10.1071/CH07168)

·       Line 324: double helices

·       Line 326: "hexagonal unit cell". The double helices are certainly not loosely arranged since it is a crystal! Please refer to proper articles.

·       Line 329: lipids would be better mentioned than alcohols because, to my knowledge, native starch granules can contain lipids but not alcohols!

·       Line 365: The SNPs obtained by acid hydrolysis of native starch granules have a higher crystallinity than the parent granules.

·       Line 370: what is a "starch level"?

·       Line 430: "showed" or "evidenced" instead of "discovered"

·       Line 435: as a function of hydrolysis time

·       Line 473: Pickering emulsions are not restricted to oil-water systems. In particular, SNPs have been used to stabilize monomer/polymer emulsions (see for instance https://doi.org/10.1016/j.jcis.2020.05.011)

·       Line 495: Nanostarch particles (not granules)

·       Line 511: SNPS from acid-hydrolyzed starch granules

·       The present Conclusion as a separate section is not useful. Please merge with the Future Research section.

Author Response

Thank you very much for your comments concerning our manuscript entitled “Starch Nanoparticles: Preparation, Properties, and Applications”. Those comments are valuable and very helpful for revising and improving our paper. We have studied the comments carefully and have made a correction which we hope meets with approval. The revised portions are marked in blue on the paper. The main correction and the responses to the reviewer’s comment are as follows:

Reviewer 3

The authors have considered most comments and corrections of the reviewers and have revised their manuscript accordingly. In particular, they mentioned that the text has been proofread by an English-speaking person. It is only partially true and a significant effort must still be done to improve the language and prevent ambiguities before publication. Some corrections and questions are indicated below and should be addressed.

Response:

We are sorry if our revised is not meet your approval. The paper has been proofread by KG Support Limited (review certification has been attached)

  • Line 32: Starch is a mixture of two macromolecules, namely…

Response:

It has been revised (Line 32)

  • Line 52: nanostarches

    Response:

    It has been revised (line 52)

  • Lines 53-54: this sentence is ambiguous. Please rephrase for english.

    Response:

    It has been revised (Lines 52-54)

  • Line 57: and which may include…

    Response: It has been revised (Lines 56-57)

  • Line 62: This review focuses on SNPs, their various…

    Response: It has been revised (Line 62)

  • Line 65: delete the first "of SNPs"

Response:

It has been deleted (Line 65)

  • Line 66: The scheme in Figure 1 summarizes the preparation, properties and applications of SNPs.

Response:

    It has been revised (Lines 66-67)

  • Line 69: Figure 1: Scheme of SNP preparation, properties and applications

Response:

It has been revised (Line 69)

  • Line 79: On the one hand, top-down methods…

Response:

It has been revised (Line 79)

  • Line 81: of large particles into small ones

Response:

It has been revised (Line 81)

  • Line 82: On the other hand, bottom-up methods…

Response:

    It has been revised (Line 82)

  • Line 85: to produce nanoparticles.

Response:

It has been revised (Line 85)

  • Line 89: I am not sure of what "sophisticated" (which is a personal qualification with which some readers might object) mean here. Equipment to fragment starch granules can also be called "sophisticated". "specific" maybe? But both approaches use specific equipment. I would rephrase anyway.

Response:

It has been revised (Line 89)

  • Line 101: Preparation Methods of Starch nanoparticles (all plural)

Response:

  It has been revised (Line 101)

  • Table 1: why don't you make a clear separation in the table between Top-down and bottom-up methods as described in the text?

Response:

It has been revised, the table have been separated in 2 sections: top-down and bottom-up methods (Table 1)

  • Table 1: be careful that most numbers contain commas! Please correct. Use the same sign for "°C". It varies too much. Define ASPU.

Response:

It has been revised (marked in blue in Table 1

Acid Stable Pullulanase Units (ASPU) is ASPU is defined as the amount of enzyme that liberates 1.0 mg glucose from starch in 1 min at pH 4.4 and 60 °C

  • Line 114: Besides, physical methods…

Response:

It has been revised (Line 114)

  • Line 114: perhaps they are less time-consuming but are they less energy-consuming (a crucial factor in the process design and industrial scale-up)?

Response:

It has been added (Lines 114-116)

  • Line 117: the breakdown of macromolecules into smaller molecules is simply called "depolymerization".

Response:

It has been revised (Line 118)

  • Line 128: The sentence "Ultrasonication destroys microwaves" does not mean anything. Please describe ultrasonication mechanism more accurately (especially because it is not related to microwave irradiation!).

Response:

It has been revised (Lines 128-137)

  • Line 136: The yield is higher because acid hydrolysis partly dissolves away the material.

Response:

It has been revised (Lines 144-147)

  • Line 141 which was applied by Naegeli…….. whereas Lintner used hydrochloric acid…

Response:

It has been revised (Lines 151-152)

  • Line 144: starch granules

Response:

It has been revised (Line 153)

  • Line 144: delete "using hydrochloric or sulfuric acid" since it has been written above.

Response:

It has been deleted

  • Line 148: No need to repeat for the third time what was written above. The reader understood that sulfuric or hydrochloric acids were used for the hydrolysis. 

Response:

It has been deleted

  • Line 151: "SNPs" instead of "starch"

Response:

It has been revised (Line 161)

  • Line 157: "sulfate groups" (according to IUPAC nomenclature)

Response:

It has been revised (Line 166)

  • Enzymatic methods: perhas t would be helpful to explain the specific action of each enzyme and why they are used to degrade starch granules.

Response:

It has been revised (Lines 171-177)

  • Line 167: This is the enzymatic section. Why do you talk about acid hydrolysis here?

Response:

Thank you for your correction. There is a typo, it should be enzymatic hydrolysis not acid hydrolysis. It has been revised (Lines 179-180)

  • Line 171: "combined methods" or "combination of methods"

Response:

It has been revised (Line 183)

  • Line 178: not "polymer"; "nanoscale particle"

Response:

It has been revised (Line 190)

  • Line 201 and rest of section 3.5: Combined methods

Response:

It has been revised (marked in blue in section 3.5)

  • Line 228: Starch Nanoparticles

Response:

It has been revised (Line 240)

  • Section 4.1: Why is only amylose content only considered and why is it particularly important? Why not amylopectin, which also influences the properties? The overall *composition*, hence the amylose/amylopectin ratio, is important. 

Response:

It has been added (Lines 247-252)

  • Line 257: please rephrase "the SNP scanning technique" that does not mean anything.

Response:

It has been revised (Lines 270-271)

  • Lines 263-264: "SNPs can be…..   The differences in morphology depend….

Response:

It has been revised (Lines 276-277)

  • Line 266: "particles", not "molecules"

Response:

It has been revised (Line 279)

  • Line 299: SNPs are spherical…..

Response:

It has been revised (Line 309)

  • Line 320: it is not exclusively ascribed. It is *mostly* ascribed. Amylose can participate to crystallites to some extent.

Response:

It has been revised (Line 329)

  • Ref. 84 is not relevant to explain the differences in the starch crystal structure. Please use proper review articles or textbook chapters, such as Perez et al. (https://doi.org/10.1002/star.201000013) or Buléon et al. (https://doi.org/10.1071/CH07168)

Response:

It has been revised (Lines 330-333 )

  • Line 324: double helices

Response:

It has been revised (Line 334)

  • Line 326: "hexagonal unit cell". The double helices are certainly not loosely arranged since it is a crystal! Please refer to proper articles.

Response:

It has been revised (Lines 335-337)

  • Line 329: lipids would be better mentioned than alcohols because, to my knowledge, native starch granules can contain lipids but not alcohols!

Response:

It has been revised (Lines 338-339)

  • Line 365: The SNPs obtained by acid hydrolysis of native starch granules have a higher crystallinity than the parent granules.

Response:

It has been revised (Lines 372-373)

  • Line 370: what is a "starch level"?

Response:

It has been revised (Lines 377-378)

  • Line 430: "showed" or "evidenced" instead of "discovered"

Response:

It has been revised (Line 435)

  • Line 435: as a function of hydrolysis time

Response:

It has been revised (Lines 440-441)

  • Line 473: Pickering emulsions are not restricted to oil-water systems. In particular, SNPs have been used to stabilize monomer/polymer emulsions (see for instance https://doi.org/10.1016/j.jcis.2020.05.011)

Response:

It has been revised (Lines 478-479)

  • Line 495: Nanostarch particles (not granules)

Response:

It has been revised (Line 501)

  • Line 511: SNPS from acid-hydrolyzed starch granules

Response:

It has been revised (Line 517)

  • The present Conclusion as a separate section is not useful. Please merge with the Future Research section.

Response:

It has been revised (Section 6, Lines 747-776)

Round 3

Reviewer 3 Report

The authors have revised their manuscript, mostly to avoid ambiguities by improving the English language. It can thus be accepted for publication. However, a careful reading of the article galley proof before final publication is strongly recommended.